

# Changing sources and burial of organic carbon in the Chukchi Sea sediments with retreating sea ice over recent centuries

Liang Su[1,2], Jian Ren[2*], Marie-Alexandrine Sicre[3], Youcheng Bai[2], Ruoshi Zhao[1,2],

Xibing Han[4], Zhongqiao Li[2], Haiyan Jin[2,5], Anatolii S. Astakhov[6], Xuefa Shi[7], Jianfang Chen[2,5*]

[1] Ocean College, Zhejiang University, Zhoushan 316021, China

[2] Key Laboratory of Marine Ecosystem Dynamics, Second Institute of Oceanography, Ministry of Natural Resources, Hangzhou 310012, China

[3] Sorbonne Université, Pierre et Marie Curie, CNRS, LOCEAN, Case 100, 4 place Jussieu, F-75005 Paris, France

[4] Key Laboratory of Submarine Geosciences, Second Institute of Oceanography, Ministry of Natural Resources, Hangzhou 310012, China

[5] State Key Laboratory of Satellite Ocean Environment Dynamics, Second Institute of Oceanography,

Ministry of Natural Resources, Hangzhou 310012, China

[6] V.I. Il'ichev Pacific Oceanological Institute, Far Eastern Branch of Russian Academy of Sciences, Vladivostok 690041, Russia

[7] Key Laboratory of Marine Geology and Metallogeny, First Institute of Oceanography, Ministry of Natural Resources, Qingdao 266061, China

*Correspondence to*: Jian Ren (jian.ren@sio.org.cn) and Jianfang Chen (jfchen@sio.org.cn)





**Abstract.** Decreasing sea ice extent in summer caused by climate change is affecting the carbon cycle
of the Arctic Ocean. In this study, surface sediments across the western Arctic Ocean are investigated to

characterize sources of sedimentary organic carbon (OC). Bulk organic parameters (total organic carbon,

total nitrogen, $\delta^{13}C_{org}$ and $\delta^{15}N$) combined with molecular organic biomarkers (e.g., sterols and highly

branched isoprenoids (HBIs)) are applied to distinguish between sympagic, pelagic, and terrestrial OC.

Furthermore, downcore profiles of these parameters were also generated from the Chukchi Sea R1 core

(74 °N) to evaluate changes in the relative contribution of these three components of sedimentary OC

over the last 200 years with decreasing sea ice. Our data evidence that from 1820s to 1930s, prevailing

high and variable sea ice cover inhibited *in situ* primary production resulting in prominent land-derived

material stored in sediments. From 1930s to 1980s, with the gradual decline of sea ice, primary

production increased progressively. The ratio of sympagic and pelagic OC began to rise to account for a

larger portion of sedimentary OC. Since 1980s, accelerated sea ice loss led to enhanced primary

production, stabilizing over the last decades due to freshwater induced surface ocean stratification in

summer.



## 1 Introduction

Knowledge on processes and feedback mechanisms controlling the carbon cycle is essential for a better understanding of the Arctic marine ecosystem dynamics and its role in climate change (Parmentier et al., 2017; Wheeler et al., 1996). The Arctic Ocean is the major world carbon sink region, where huge amounts of marine and terrestrial organic carbon (OC) have been accumulated (Stein et al., 2004). Today, the Arctic Ocean experiences unprecedented changes caused by global warming and Arctic amplification (Cavalieri et al., 1997; Rantanen et al., 2022; Serreze and Francis, 2006; Shindell and Faluvegi, 2009) which have resulted in major sea ice loss with consequences on the marine ecosystems and Arctic carbon budget. Increased river discharge and melting permafrost are responsible for enhanced delivery of terrigenous inorganic and organic carbon to the Arctic marginal seas (Grotheer et al., 2020; Holmes et al., 2011; Rawlins et al., 2021; Vonk et al., 2012). Terrigenous OC reaching the Arctic Ocean is either partly mineralized or transported to the sea floor where it is ultimately buried (Fritz et al., 2017; Tanski et al., 2019). Increased nutrient-rich waters brought by enhanced Pacific Water Inflow (PWI), a major source of nutrients to the Arctic, also contributed to stimulate phytoplankton productivity in the western Arctic Ocean (Arrigo and van Dijken, 2015; Tian et al., 2021; Woodgate and Peralta-Ferriz, 2021; Woodgate, 2018). Lastly and most importantly, the rapid sea ice loss in summer has resulted in large areas of the Arctic Ocean that shifted from multi-years to seasonal sea ice coverage (Cavalieri and Parkinson, 2012; Parkinson et al., 1999; Stroeve et al., 2007) allowing higher light penetration in surface waters thereby enhancing primary production and export of OC to the bottom floor. Enhanced summer sea ice melting further contributes to sea ice algal production, export and burial of marine OC in sediments (Ardyna and Arrigo, 2020). The Chukchi Sea (CS) is one of the most productive regions of the Arctic marginal seas (Cai et al., 2010; Ouyang et al., 2022; Zhuang et al., 2022). With the rapid sea ice retreat, the CS is becoming a key area to study climate induced OC cycle changes since the beginning of the Industrial Era.

A large variety of indicators, including bulk geochemical ratios and lipid biomarkers have been developed to characterize the composition of OC in Arctic Ocean sediments (Volkman, 1986; Fernandes and Sicre, 2000; Sparkes et al., 2015). Among them, lignins and $\delta^{13}C_{org}$ have been successfully used to provide reliable estimates of terrestrial OC (Tesi et al., 2014; Wang et al., 2019a; Wild et al., 2022).



65 However, pelagic and sympagic sourced OC remains difficult to discriminate in the Arctic Ocean,

particularly in regions of high sympagic productivity. To address this issue, the H-print index based on

highly-branched isoprenoids (HBIs) defined as the ratio of pelagic HBI-III over the sum of sympagic

($IP_{25}$ and HBI-II) and pelagic biomarkers (HBI-III) was developed (Brown et al., 2014b; Koch et al.,

2020). Values close to 100% are thus indicative of prominent pelagic sources while those close to 0%

70 reflect prevailing sympagic sources. By combining H-print and $\delta^{13}C_{org}$ we here intend to more accurately

quantify marine pelagic, marine sympagic and terrestrial fractions of OC in Arctic sediments.

 Documenting changes in sea ice changes and induced transformations of Arctic ecosystems is key

to better predict how the carbon cycle will respond to future changes with continued warming (Arrigo et

al., 2008; Bates and Mathis 2009). Extensive surveys of Arctic sea ice have been possible only since the

75 1970s due to the development of remote sensing observations (Cavalieri et al., 1996). Prior to this, very

few *in situ* observations on sea ice exists due to the inaccessibility of the Arctic Ocean. Paleoclimate

proxies such as micropaleontological fossils and geochemical indicators have thus been used as

alternative approaches to document past changes of sea ice and place them in the context of ongoing

changes (e.g. Belt et al., 2007; de Vernal et al., 2013). The monounsaturated HBI biomarker $IP_{25}$ (Ice

80 Proxy with 25 carbon atoms) produced by sea ice diatoms was initially proposed to assess seasonal sea

ice cover (Belt et al., 2007; Masséet al., 2008). The $PIP_{25}$ (Phytoplankton-$IP_{25}$) index, that combines $IP_{25}$

with pelagic phytoplankton biomarkers, was then proposed to provide semi-quantitative estimates of

seasonal sea ice (Belt, 2019; Müller et al., 2011). Most $IP_{25}$-related studies in the Arctic Ocean have

focused on surface sediments to derive spatial seasonal sea ice distribution (Kolling et al., 2020; Stoynova

85 et al., 2013; Su et al., 2022; Xiao et al., 2015a; Xiao et al., 2013) or its past variability at the millennial

scale and beyond (Cronin et al., 2013; Polyak et al., 2016; Stein et al., 2017; Xiao et al., 2015b), but only

a limited number has explored sea ice variability over the past centuries (Bai et al., 2022; Hu et al., 2020;

Kim et al., 2019). None have attempted to link seasonal sea ice changes to sedimentary OC composition

since the beginning of the Industrial Era.

90 In this study, we investigate the potential of H-print combined with $\delta^{13}C_{org}$ in surface sediments of

the northern CS to discriminate and quantify the relative contribution of OC originating from pelagic,

sympagic and terrestrial sources and their evolution over the last two centuries under changing sea ice



conditions to improve our understanding of ongoing alteration of the OC cycle.

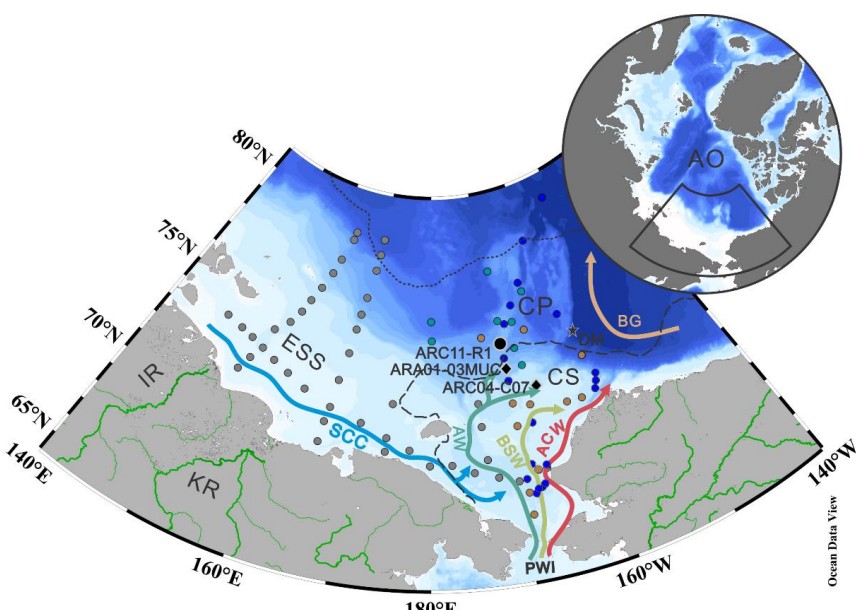

**Figure 1**: Map of the western Arctic Ocean (AO) showing surface ocean circulation and sampling locations (black, gray, and green dots). The stations in gray and green represent the surface samples collected on the LV77 and 11th CHINARE cruises, respectively. The black dot represents the sediment core ARC11-R1. The black diamonds show other sediment cores discussed in the text (ARC04-C07, Bai et al 2022; ARA01-03MUC, Kim et al 2019). Surface sediments reported by Bai et al. (2019) and Wang et al. (2017) were marked in blue and orange dots, respectively. The black pentagram represents the time series sediment trap DM station (Bai et al., 2019). The dotted and dashed lines in black represent the 20% isolines of September sea-ice concentration for the 2019 and 1979, respectively. Main study regions: ESS, East Siberian Sea; CS, Chukchi Sea; CP, Chukchi Plateau. Surface circulation: SCC, Siberian coastal current; PWI, Pacific water inflow; ACW-Alaskan Coastal Water; AW-Anadyr Water; BSW-Bering Shelf Water. Rivers are shown in green lines: IR, Indigirka River; KR, Kolyma River.

## 2 Oceanographic setting

The CS is one of the largest marginal seas in the world located on the northern Asian and American continents (Jakobsson, 2002). The surface ocean circulation in the CS is controlled by winds and sea ice cover (Ovall et al., 2021). This basin is connected to the Pacific Ocean through the Bering Strait where the PWI entering the Arctic Ocean strongly influences the physico-chemical water properties of the



Arctic Ocean and contribute to enhance primary production (Coachman and Aagaard, 1966) (Fig. 1). In
the CS, the PWI divides into three branches: the highly saline and high-nutrient content Anadyr Water
(AW) on the western side, the fresher and oligotrophic Alaska Coastal Water (ACW) on the eastern side,
and the moderately saline Bering Shelf Water (BSW) in between (Grebmeier et al., 2006; Hunt et al.,
2013; Woodgate et al., 2005). The eastern side of the CS is adjacent to the Beaufort Sea. The dynamics
of the Beaufort Gyre (BG) also impacts on the characteristics of the CS water mass. In particular,
enhanced anticyclonic BG circulation has resulted in increased freshwater convergence into the Canadian
Basin in recent years (Giles et al., 2012) with implications on the local biological production as well as
on the transport of terrestrial organic matter (He et al., 2012; Coupel et al., 2015; Ren et al., 2020). Fresh
and cold waters from the seasonal Siberian Coastal Current (SCC) is another feature of the surface ocean
circulation that lowers salinity of the central CS water (Weingartner et al., 1999). The dramatic loss of
seasonal sea ice is summer caused by global warming is particularly tangible in the Arctic Ocean
marginal seas (Cavalieri et al., 1997; Parkinson et al., 1999; Polyakov et al., 2003; Zhang et al., 2021)
and most pronounced in the CS both in terms of sea ice extent and thickness (Serreze and Stroeve 2015;
Wang et al., 2019b). Remote sensing data evidence strong seasonal variations, with sea ice being present
in the CS from November to June, receding as the summer season (July) begins until minimum extent is
reached in September (https://nsidc.org, Cavalieri et al., 1996).

## 3 Material and methods

### 3.1 Material

A total of 42 surface sediments (0-2 cm) collected across the East Siberian Sea (ESS) and CS were
recovered during the Cruise LV77 aboard the R/V *Akademik M.A. Lavrentiev*. Additional 11 surface
sediments (0-2 cm) and a 15 cm long sediment core ARC11-R1 (R1 hereafter, 74.64° N, 169.13° W, 200
m water depth) were also collected using a box corer and a multi-corer, respectively, in the CS and
Chukchi Plateau (CP) during the 11[th] Chinese National Arctic Research Expedition (CHINARE) in
summer 2020 aboard the R/V *Xuelong 2* (Fig. 1). Subsampling was performed on board at a sampling
interval of 1 cm. Subsampled core sediments and surface sediments were quickly frozen after recovery
at -20 °C until further analysis in the laboratory.

### 3.2 Core chronology



The chronology of the R1 core is based on $^{210}Pb_{ex}$ determination measurements performed at the State Key Laboratory of Estuarine and Coastal Research, East China Normal University, Shanghai, China, using an HPGe gamma spectrometry (GSW275L, Canberra). The excess $^{210}Pb$ ($^{210}Pb_{ex}$) activity was

calculated by subtracting the supporting fraction ($^{226}Ra$) from the total $^{210}Pb$ ($^{210}Pb_{total}$) activity in the sediment. The error in $^{210}Pb_{ex}$ is computed by propagating the error in the corresponding measured pair ($^{210}Pb$ and $^{226}Ra$). A mean linear sedimentation rate (cm yr$^{-1}$) was calculated from the $^{210}Pb_{ex}$ profile with depth in sediment using a Constant Flux-Constant Sedimentation Rate (CF-CS) model, assuming continuous homogeneous deposition of non-equilibrium $^{210}Pb$ in the sediment (Nittrouer et al., 1984). As

$^{137}Cs$ was also measured during the same gamma counting session, the onset of its activities in the sediment was used to test the chronology.

### 3.3 Bulk analyses

Total organic carbon (TOC), total nitrogen (TN), $\delta^{13}C_{org}$, and $\delta^{15}N$ of 42 surface sediments from the ESS as well as 11 surface sediments and the R1 core from the CS were analyzed at the Key Laboratory

of Marine Ecosystem Dynamics, Second Institute of Oceanography, Ministry of Natural Resources (MED, SIO, MNR, Hangzhou, China). There were first freeze-dried and then ground and homogenized before bulk parameter analyses. About 0.5 g of sediment was acidified using 1 mol L$^{-1}$ HCl and heated overnight in a water bath at 50 °C. The excess acid was washed away using ultrapure water (Williford et al., 2007). These samples were weighed for TOC and $\delta^{13}C_{org}$ determination. For TN and stable isotopes

of nitrogen ($\delta^{15}N$) analyses, we used samples that were not acidified. TOC, TN, $\delta^{13}C_{org}$, and $\delta^{15}N$ measurements were carried out on an elemental analyzer (EA, Elementar CHNOS) coupled to an isotope ratio mass spectrometer (IRMS, Thermo, Delta V advantage). The standard deviations for TOC, TN, $\delta^{13}C_{org}$, and $\delta^{15}N$ based on replicate analyses were 0.02%, 0.005%, 0.2‰, and 0.2‰, respectively.

### 3.4 Biomarker Analyses

Biomarker analyses were completed at MED, SIO, MNR (Hangzhou, China). Before extraction, internal standards 7-hexylnonadecane and cholest-5-en-3β-ol-D6 were added to about 5 g of freeze-dried and homogenized sediment for quantification of HBIs and sterols, respectively. Extraction was performed 3 times in an ultrasonic bath for 15 min using dichloromethane/methanol (2:1 v/v). The 3 extracts were combined and dried under a gentle nitrogen stream. Further separation was carried out by



adsorption chromatography on an open-column filled with $SiO_2$ using 2.5 ml n-hexane and 4 ml n-hexane/ethyl acetate (70:30 v/v) to separate the hydrocarbons and sterol, respectively, from the total lipid extract. About 50 μl BSTFA (bis-trimethylsilyl-trifluoroacetamide) were added to the sterol fraction and heated at 70 °C for 1 hr for silylation.

Then, both hydrocarbons and sterols were analyzed by gas chromatography (GC, Agilent

Technologies 7890, 30 m HP-1MS column, 0.25 mm in diameter, and 0.25 μm film thickness) coupled to mass spectrometry (MS, Agilent 262 Technologies 5975C inert XL). A heating rate of 10 °C min$^{-1}$ for the oven temperature was programmed from 40 °C to 300 °C and maintained at final temperature for 10 min. The ion source temperature was set at 250 °C and ionization energy at 70 eV for MS analyses (Belt et al., 2007, Müller et al., 2009). Individual compounds were identified based on their retention time and

mass spectra. Selective ion monitoring was used to detect the $C_{25}$-HBIs ($m/z$ 350 for $IP_{25}$, $m/z$ 348 for HBI-II, and $m/z$ 346 for HBI-III) and the sterols ($m/z$ 470 for brassicasterol (24-methylcholesta-5,22E-dien-3β-ol), $m/z$ 500 for dinosterol (4α,23,24R-trimethyl-5α-cholest-22E-en-3β-ol), $m/z$ 396 for β-sitosterol (24-ethylcholest-5-en-3-ol) and $m/z$ 382 for campesterol (24-methylcholest-5-en-3β-ol). Concentrations of HBI were determined based on the area of individual compounds and that of the 7-

hexylnonadecane ($m/z$ 266) obtained by GC/MS. Similarly, sterol concentrations were calculated from the area of individual sterols and cholesterol-d6 ($m/z$ 464) (Belt et al., 2012, Müller et al., 2011). Concentrations of all biomarkers were normalized to TOC.

**3.5 $PIP_{25}$ Index and H-print**

$PIP_{25}$ indexes were calculated to estimate seasonal sea ice concentrations (Müller et al., 2011) using

the following expression:

$$PIP_{25} = \frac{[IP_{25}]}{[IP_{25}] + [phytoplankton\ biomarker] * c} \times 100\% \qquad (1)$$

$$where \quad c = \frac{mean\ IP_{25}\ concentration}{mean\ phytoplankton\ biomarker\ concentration} \qquad (2)$$

Brassicasterol (B), dinosterol (D), and HBI-III (III) were used as a reference for pelagic phytoplankton to calculate the $P_B IP_{25}$, $P_D IP_{25}$, and $P_{III} IP_{25}$ values, respectively.




$$H - print\% = \frac{[HBI - III]}{[IP_{25}] + [HBI - II] + [HBI - III]} \times 100 \qquad (3)$$

The H-print values were also calculated to infer the relative contribution of pelagic and sympagic OC (Brown et al., 2014b, Koch et al., 2020). Low H-print values are indicative of higher sympagic production while high H-print values point to prevalent pelagic algae.

**3.6 Environmental dataset of surface sediment**

Assessment of seasonal sea ice spatial distribution is based on the compilation of previously published surface sediment data from the ESS (HBIs, Su et al., 2022) and the CS (HBIs, Bai et al., 2019; $\delta^{13}C_{org}$, Wang et al., 2017) and new data from the CS and CP produced in this study. $\delta^{13}C_{org}$ data from the ESS are also new together with all data from the R1 core.

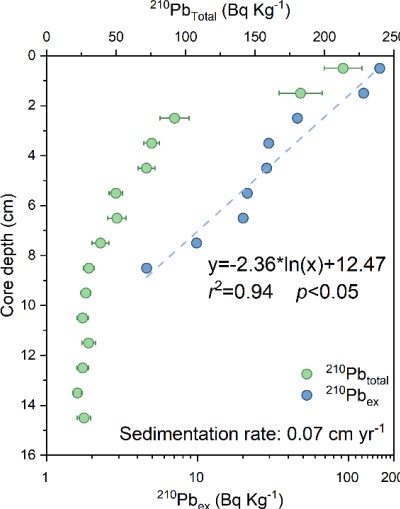

**Figure 2:** Downcore profile of $^{210}Pb_{total}$ (green circles) with the error range and $^{210}Pb_{ex}$ (blue circles) for ARC11-R1.

**4 Results**

**4.1 Chronology of ARC11-R1**

The activity of $^{210}Pb_{total}$ ranges from 22 to 214 Bq Kg$^{-1}$, with an average value of 64.30 Bq Kg$^{-1}$ (Fig. 2). $^{210}Pb_{ex}$ decreases exponentially with increasing depth and reaches negligible values around 9 cm. The calculated average sedimentation rate of R1 using the CF-CS model is estimated to 0.07 cm yr$^{-1}$ ($r^2$=0.94, $p$<0.05), which gives a time span of about 200 years for the whole core. The $^{137}Cs$ profile (not shown)



**Table 1.** Summary of bulk parameters and biomarker data from surface sediment in the Chukchi Sea and Chukchi Plateau.

| Station | Long. | Lat. | TOC (wt%) | TN (wt%) | $\delta^{13}C_{org}$ (‰) | $\delta^{15}N$ (‰) | IP$_{25}$ relative abundance* | HBI-II relative abundance* | HBI-III relative abundance* | Brassicasterol* | Dinosterol* | Campesterol+ β-sitosterol* |
|---|---|---|---|---|---|---|---|---|---|---|---|---|
| E1 | -179.89 | 75.01 | 1.23 | 0.17 | -23.72 | 9.42 | 1.58 | 1.23 | 0.17 | 25.34 | 3.11 | 189.29 |
| P2-5 | -163.68 | 76.60 | 0.87 | 0.12 | -22.73 | 8.67 | 0.22 | 0.18 | 0.04 | 1.79 | 0.60 | 37.33 |
| R5 | -168.94 | 77.76 | 1.18 | 0.15 | -23.54 | 9.05 | 0.52 | n.d. | n.d. | 3.05 | 0.92 | 44.19 |
| E2 | 179.99 | 75.84 | 0.57 | 0.13 | -22.99 | 8.92 | 3.90 | n.d. | n.d. | 20.07 | 3.59 | 127.45 |
| R2 | -168.92 | 75.61 | 0.96 | 0.18 | -22.64 | 9.91 | 1.00 | 0.69 | 0.76 | 19.30 | 4.28 | 135.13 |
| P3-7 | -165.92 | 78.61 | 0.65 | 0.12 | -22.26 | 8.01 | n.d. | n.d. | n.d. | 3.27 | 0.65 | 63.21 |
| P1-6 | -166.62 | 75.44 | 0.69 | 0.16 | -23.13 | 9.44 | 1.00 | 0.59 | 1.47 | 7.85 | 2.81 | 86.21 |
| Z4 | -166.61 | 73.54 | 1.42 | 0.28 | -22.89 | 8.74 | 5.49 | 6.52 | 3.06 | 21.89 | 7.04 | 141.89 |
| Z3 | -167.16 | 74.34 | 0.78 | 0.21 | -22.52 | 9.41 | 4.45 | 4.32 | 2.31 | 97.67 | 24.09 | 528.94 |
| P3-8 | -162.58 | 78.36 | 0.61 | 0.12 | -22.94 | 8.04 | n.d. | n.d. | n.d. | 5.56 | 1.02 | 79.41 |

* in µg g$^{-1}$ TOC

**Table 2.** Summary of bulk parameters and biomarker data from core ARC11-R1.

| Core depth (cm) | Age (yr AD) | TOC (wt%) | TN (wt%) | C/N Ratio | $\delta^{13}Corg$ (‰) | $\delta^{15}N$ (‰) | IP$_{25}$ Relative abundanc* | HBI-II Relative abundance* | HBI-III Relative abundance* | Brassicasterol* | Dinosterol* | Campesterol+ β-sitosterol* |
|---|---|---|---|---|---|---|---|---|---|---|---|---|
| 0-1 | 2013 | 1.18 | 0.18 | 7.76 | -23.51 | 9.35 | 0.45 | 0.77 | 0.51 | 40.04 | 8.73 | 86.00 |
| 1-2 | 2000 | 1.13 | 0.17 | 7.66 | -23.79 | 9.32 | 0.66 | 0.88 | 0.62 | 34.25 | 5.63 | 69.22 |
| 2-3 | 1986 | 0.94 | 0.15 | 7.15 | -23.46 | 9.05 | 0.82 | 0.93 | 0.47 | 33.61 | 5.09 | 78.85 |
| 3-4 | 1972 | 0.95 | 0.13 | 8.39 | -24.40 | 8.48 | 0.38 | 0.53 | 0.23 | 21.25 | 3.16 | 52.94 |
| 4-5 | 1959 | 0.73 | 0.12 | 7.27 | -24.17 | 7.66 | 0.22 | 0.44 | 0.16 | 16.78 | 3.53 | 36.75 |
| 5-6 | 1945 | 0.77 | 0.11 | 7.94 | -24.25 | 7.53 | 0.24 | 0.43 | 0.16 | 14.58 | 3.03 | 28.39 |
| 6-7 | 1932 | 0.81 | 0.11 | 8.62 | -24.27 | 7.17 | 0.85 | 0.71 | 0.23 | 14.88 | 3.00 | 35.05 |
| 7-8 | 1918 | 0.77 | 0.11 | 8.41 | -24.30 | 6.56 | 0.39 | 0.74 | 0.31 | 15.18 | 2.75 | 33.48 |
| 8-9 | 1904 | 0.85 | 0.10 | 9.49 | -24.63 | 6.22 | 0.23 | 0.75 | 0.40 | 16.02 | 2.76 | 23.36 |
| 9-10 | 1891 | 0.81 | 0.11 | 8.60 | -24.63 | 6.64 | 0.24 | 0.98 | 0.26 | 14.64 | 3.31 | 20.77 |
| 10-11 | 1877 | 0.84 | 0.11 | 8.75 | -24.87 | 6.67 | 0.21 | 0.93 | 0.27 | 13.14 | 3.19 | 30.66 |
| 11-12 | 1864 | 0.86 | 0.11 | 8.96 | -24.76 | 6.74 | 0.18 | 1.01 | 0.38 | 13.03 | 2.69 | 23.97 |
| 12-13 | 1850 | 0.79 | 0.11 | 8.59 | -24.93 | 6.38 | 0.19 | 0.94 | 0.42 | 12.37 | 2.78 | 16.89 |
| 13-14 | 1837 | 0.73 | 0.09 | 9.12 | -25.19 | 6.31 | 0.19 | 0.63 | 0.20 | 8.77 | 2.46 | 12.02 |
| 14-15 | 1823 | 0.71 | 0.09 | 9.16 | -25.07 | 5.43 | 0.77 | 0.21 | 0.30 | 6.51 | 2.40 | 14.62 |

* in µg g$^{-1}$ TOC

further supports the $^{210}$Pb dating. This value falls within the range reported by Cooper and Grebmeier (2018) in a Chukchi Shelf core (0.03-0.37 cm yr$^{-1}$) and is slightly lower than found at the ARC4-C07 core (0.09 cm yr$^{-1}$, Bai et al., 2022) and ARA01B-03MUC core (0.09 cm yr$^{-1}$, Kim et al., 2019) both





located to the south (Fig. 1).

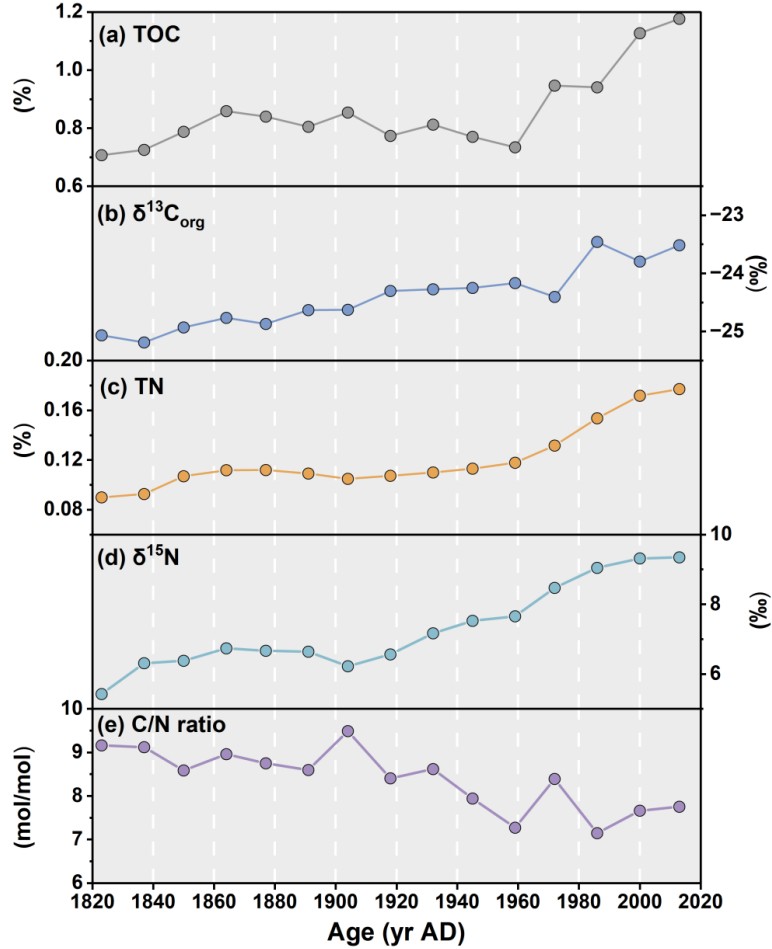

**Figure 3:** Downcore profiles of (a) Total Organic Carbon, TOC in %, (b) Organic carbon isotopic composition

($\delta^{13}C_{org}$) of the TOC in ‰, (c) Total Nitrogen, TN in %, (d) nitrogen isotopic composition ($\delta^{15}N$) in‰, and (e) C/N

ratio in the ARC11-R1 core.

**4.2 Proxy data**

*4.2.1 Surface sediment*

        The TOC and TN of surface sediments range from 0.57% to 1.42% and from 0.12% to 0.28%

respectively (Table 1, Fig. A1). $\delta^{13}C_{org}$ vary from -23.7‰ to -22.0‰ and $\delta^{15}N$ values from 8.01‰ to




9.91‰. Both HBIs and pelagic phytosterol concentrations showed a gradual decrease from the shelf to

the northern CP. The concentrations of HBI-II and HBI-III reached their detection limit at around 76 °N,

whereas for $IP_{25}$ this limit is achieved north of 78 °N. By contrast, brassicasterol and dinosterol were

detected in all samples with highest values recorded at the shelf edge. Terrestrial sterols (β-sitosterol

and campesterol) showed high values over the shelf and minimum ones at the northern end of the CP.

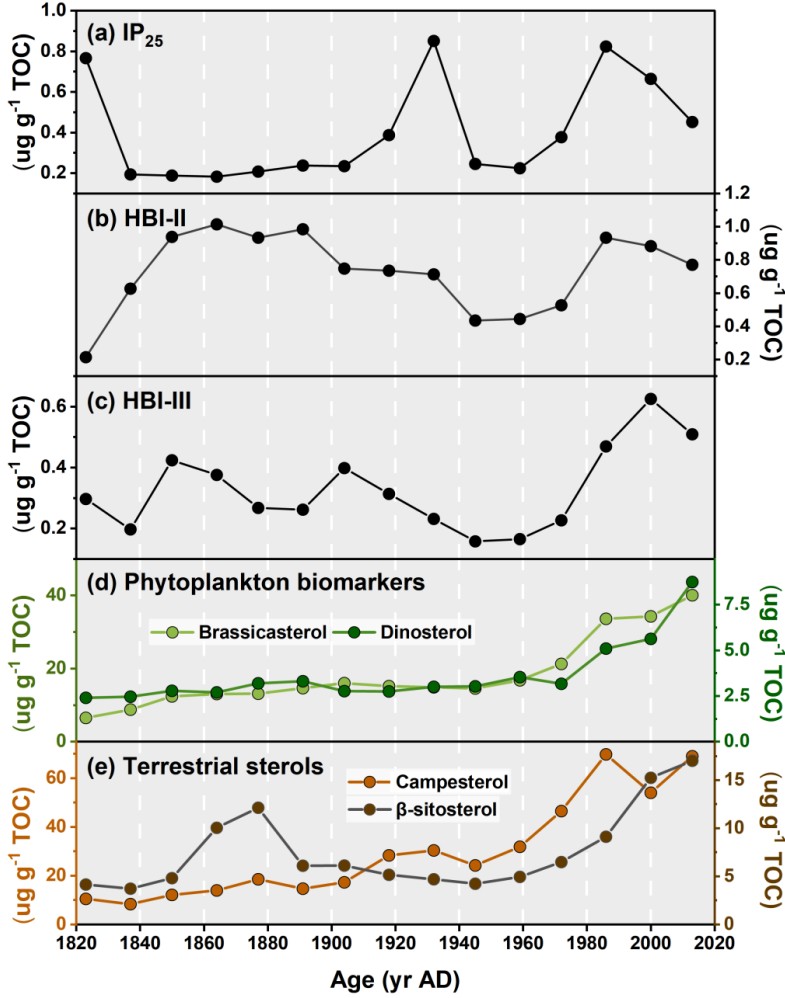

**Figure 4:** Downcore profiles of the concentrations of (a) $IP_{25}$, (b) HBI-II, (c) HBI-III, (d) brassicasterol and

dinosterol, and (e) campesterol and β-sitosterol in the ARC11-R1 core.



### *4.2.2 ARC11-R1 core*

The TOC and TN downcore profiles over the last 200 years both show increasing values towards Present ($r^2$=0.88, $p$<0.01) with values varying from 0.71% to 1.18% and from 0.09% to 0.18%, respectively (Table 2, Fig. 3a and c). TOC exhibits a minimum end of the 1950s and rapidly increase thereafter. Downcore values of C/N ratios show a gradual decrease from 9.2 to 7.6 while $\delta^{13}C_{org}$ consistently increase from -25.07‰ to -23.46‰ in the 1980s (Table 2, Fig. 3b and d). $\delta^{15}N$ exhibit constant values until the early 1900s after which they gradually increase and reach a maximum at the very top of the core (9.35‰; Table 2, Fig. 3d).

The $IP_{25}$ span from 0.18 to 0.85 µg g$^{-1}$ TOC with highest values found in the 1930s and after the1980s (Fig. 4a). Brassicasterol and dinosterol display similar patterns (Fig. 4d, $r^2$=0.85, $p$<0.01) both showing upward trends towards the present, brassicasterol being significantly more abundant (brassicasterol: 6.51 µg g$^{-1}$ TOC to 40.04 µg g$^{-1}$ TOC, dinosterol: 2.40 µg g$^{-1}$ TOC to 8.73 µg g$^{-1}$ TOC; Fig. 4d). Finally, terrestrial sterols slowly increase from 1820s to present in R1 core (campesterol: 10.47 µg g$^{-1}$ TOC to 68.98 µg g$^{-1}$ TOC, β-sitosterol: 4.15 µg g$^{-1}$ TOC to 17.02 µg g$^{-1}$ TOC; Fig. 4e).

### 5 Discussion

#### 5.1 Reconstruction of sea ice condition

Sympagic biomarker $IP_{25}$ concentrations (0.40 ± 0.25 µg g$^{-1}$ TOC, mean ± S.D.) were lower throughout the R1 core than found over same period in the ARA01B-03MUC (0.96±0.72 µg g$^{-1}$ TOC, Kim et al., 2019) and ARC04-C07cores (1.29±1.19 µg g$^{-1}$ TOC, Bai et al., 2022) both located south of our core (Fig. A2). Decreasing sympagic biomarker concentrations with increasing latitude likely reflect lower export to the sea floor due to icier conditions in the North. The box-plot also points to higher decadal variability at the two southernmost sites possibly reflecting sea ice edge (Fig. A2). The presence of $IP_{25}$ throughout R1 indicates that sea ice cover has been seasonal at least since the 1820s at this location. While Bai et al. (2022) reported parallel trends of $IP_{25}$ and HBI-II throughout the ARC4-C07 core, this feature is only observed since the 1930s in core R1 (Fig. 4a and b). Indeed, before 1930s, these two HBIs show opposite behavior. The reasons for this discrepancy are not entirely clear and few data exist to explore in depth possible interpretations. The shift of HBI-II/$IP_{25}$ from high values (3.3 to 5.6, except for 1823) to lower ones (0.8 to 2.0) in our core around 1918 could witness different sea ice conditions as



hypothesized by Cabedo-Sanz et al. (2013). Indeed, enhanced HBI-II/IP$_{25}$ was reported under more variable sea ice conditions caused by warmer settings (Belt et al., 2007; Cabedo-Sanz et al., 2013), which

is supported by PIP$_{25}$ inferred sea ice reconstruction along the core (see discussion below). Another possible explanation involves a change in HBI producers. HBI-II is also found in Southern Ocean sediments, unlike IP$_{25}$, where its production has been attributed to the sea ice diatom *Berkeleya adeliensis* Medlin (Belt et al., 2016; Brown et al., 2014a). It was also noted that *B. adeliensis* tends to preferentially flourish in platelet ice, particularly in coastal settings, leading to link its occurrence to landfast sea ice

associated with freshwater discharge in Southern Ocean sediments (Belt, 2018, 2019). The divergent behavior the HBI-II and IP$_{25}$ prior 1918 could thus be indicative of variable sea ice conditions with a possible contribution of drifting ice from coastal areas of the ESS. Highest mean HBI-II/IP$_{25}$ value of 2.9 found North of Iceland under prevailing drifting ice influence is in favor of this hypothesis (Massé et al., 2008).

Combined IP$_{25}$ and pelagic phytoplankton biomarkers were investigated to quantify seasonal sea ice cover. P$_B$IP$_{25}$, P$_D$IP$_{25,}$ and P$_{III}$IP$_{25}$ were calculated using surface sediment balance factors *c* of 0.02, 0.11 (Xiao et al., 2015a) and 0.63 (Smik et al., 2016), respectively. All indexes show similar trends reflecting the strong correlation between brassicasterol, dinosterol and HBI-III (Fig. 5 and A3; all: *r*>0.63, *p*<0.01). Only in the 1820s and in the 1930s were PIP$_{25}$ values above threshold off permanent sea ice (75%, Müller

et al., 2011) (Fig. 5). Most PIP$_{25}$ values were otherwise distributed between 30% and 60% sea ice cover.

Between the 1820s and 1850s, all PIP$_{25}$ values steeply drop featuring a rapid sea ice retreat. Then, P$_B$IP$_{25}$ and P$_D$IP$_{25}$ show rather stable values until the beginning of the 20th century whereas P$_{III}$IP$_{25}$ slowly increase (Fig. 5). All three PIP$_{25}$ indexes point to seasonal sea ice or marginal ice zone conditions. Higher amounts of HBI-III and P$_{III}$IP$_{25}$ indicate sea ice edge bloom conditions. Furthermore, sediment trap data

at the DM station (74 °N) evidenced high levels of brassicasterol till late summer/early autumn when ice free conditions were reached whereas HBI-III was close to the detection limit mid-summer when sea ice has melted (Bai et al., 2019, Gal et al., 2022). This result further confirm that HBI-III producers proliferate at the sea ice edge rather than in ice free waters. From 1850s to 1910s, both P$_B$IP$_{25}$ and P$_D$IP$_{25}$ were below 50%, pointing to less and variable sea ice cover than end of early 19th century. HBI-III falls

in a high value of sea ice cover range reflecting icier conditions in the 1820s and 1910s. However, low



concentrations of $IP_{25}$ and phytoplankton biomarkers during this period point to permanent sea ice cover

(Fig. 6a, b, and c), which may bias the application of the $PIP_{25}$ index. Highest values of $PIP_{25}$ in the 1930s

further support heavy sea ice likely associated with colder conditions. Therefore, the permanent sea ice

appeared in the western CS in the between the 1820s to 1930s.

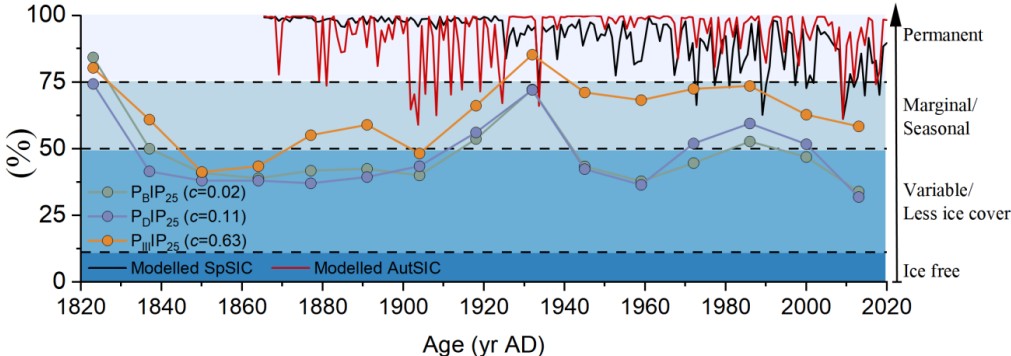

**Figure 5:** $PIP_{25}$ index using brassicasterol (in cyan, $P_BIP_{25}$, c = 0.02 as calculated by Xiao et al (2015)), dinosterol
(in purple, $P_DIP_{25}$, c = 0.11 as calculated by Xiao et al (2015)), and HBI-III (in orange, $P_{III}IP_{25}$, c = 0.63 as calculated
by Smik et al (2016)) and modelled sea ice concentration based on CMIP6 by Wu et al (2019) (in black, SpSIC; in
red, AutSIC). The degree of the sea ice conditions from low to high level is in the order of ice-free, ice edge, extended
ice cover, and permanent ice cover.

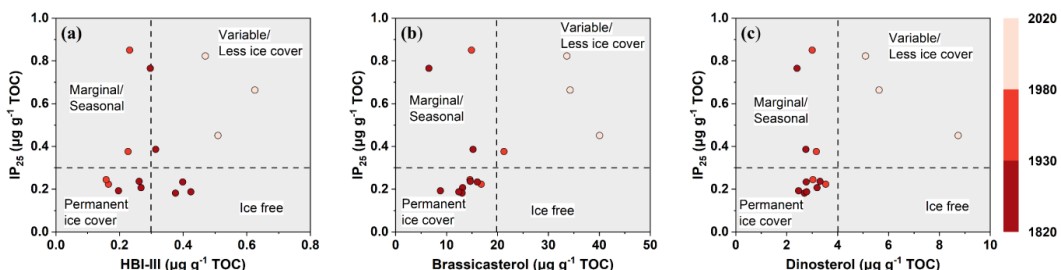

**Figure 6:** Correlations between the concentrations of $IP_{25}$ and phytoplankton biomarkers ((a) HBI-III, (b)
brassicasterol and (c) dinosterol, respectively) in the ARC11-R1 core to define different spring/summer sea ice
condition zones. The gradient from dark to light red represents periods of 1820-1930, 1930-1980, and 1980-2020,
respectively.

After 1930s, $P_{III}IP_{25}$ gradually decrease to ca 70% while $P_BIP_{25}$ and $P_DIP_{25}$ drop to lower values

around 40% till the 1960s to then slightly increased until the 80s -90s when $P_BIP_{25}$ and $P_DIP_{25}$ exceed the

seasonal sea ice threshold value (50%) and $P_{III}IP_{25}$ that of nearly permanent sea ice at approximately 75%.

In this time interval (1930s - 1980s), low $IP_{25}$ and increasing brassicasterol in cores ARC4-C07 and

ARA01B-03MUC suggest enhanced sea ice melting and the northward retreat of the summer ice edge

(Fig. 7 and A4). From 1980s to present, $P_BIP_{25}$ and $P_DIP_{25}$ continued to decrease but at a faster rate (Fig.





5 and 7) emphasizing the unprecedented decline of seasonal sea ice over the last 30 years, as

reconstructed by geochemical proxy in the sediment core (Astakhov et al., 2019) and documented by

remote sensing data (Walsh et al., 2019, Wang et al., 2019b).

In summary, the downcore profiles of seasonal sea ice proxies over the last 200 years evidence

(Fig. 7d): i) nearly permanent sea ice between the 1820s and 1930s; ii) from 1930s to 1980s, the seasonal

sea ice slowly retreated to the north and the summer sea ice edge gradually approached the location of

the R1 core; iii) a strong reduction sea ice cover with summer sea ice edge conditions since the 1980s.

Finally, our findings highlight the value to further investigate the potential of HBI-II in the Arctic Ocean

to track drifting and landfast ice from coastal regions under freshwater discharge influence.

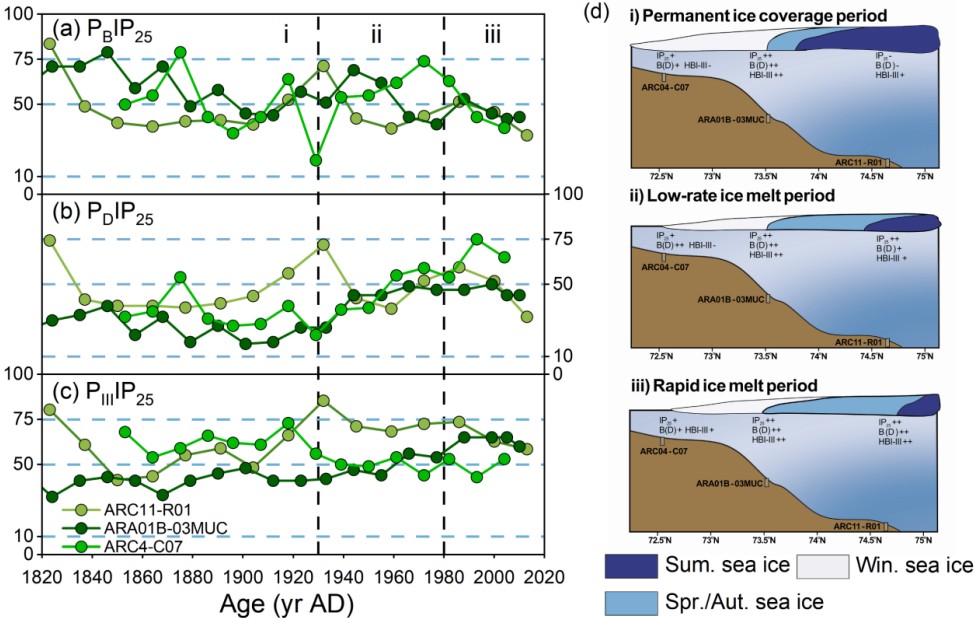

**Figure 7:** Downcore biomarker profiles of (a) PIP$_{25}$ index using brassicasterol (P$_B$IP$_{25}$), (b) dinosterol (P$_D$IP$_{25}$) and
(c) HBI-III (P$_{III}$IP$_{25}$) in the core ARC04-C07 (Bai et al. 2022), ARA01B-03MUC (Kim et al. 2019) and ARC11-R1.
i, ii and iii in (a), (b) and (c) represent different sea ice cover scenarios, which are schematically illustrated in (d).
The color of sea ice in (d) represents the different sea ice cover: white for winter sea ice, light blue for spring/autumn
sea ice, and dark blue for summer sea ice.

**5.2 Organic carbon variability in response to sea ice change**

*5.2.1 Modern sources of organic carbon*

The loss of sea ice is the most remarkable manifestation of global warming in the Arctic that has



profound impacts on the carbon cycle. The most obvious one is the shift of the primary production pattern

as a result of light and nutrient supply changes. Enhanced riverine inputs of terrestrial organic matter also

affect the nature and amount of organic carbon reaching the Arctic Ocean. In this section, we investigate

OC compositional changes over the past 200 years in relation with sea ice conditions by distinguishing

three major carbon pools, e.g. sympagic, pelagic and terrestrial. We use the H-print ratio as a

complementary indicator to $PIP_{25}$ to discriminate pelagic and sympagic marine sources (Brown et al.,

2014b; Brown and Belt, 2017; Koch et al., 2020). As shown in figure 8a, high values of H-print (>80%)

occur in the western CS where marine phytosterols are high (dinosterol concentrations >5 µg g$^{-1}$ TOC

and brassicasterol concentrations >100 µg g$^{-1}$ TOC, Su et al., 2022). Intermediate values (30%-60%) are

roughly lying between the 1979 and 2019 isolines of the September minimum ice edge, denoting mixed

pelagic and sympagic productions (Fig 8a). Minimum H-print values are found in coastal sediments along

the western ESS and suggest freshwater discharge by lowering salinity waters may have suppressed HBI-

III production (Su et al., 2022) (Fig. 8a). At higher latitudes, low H-print values are likely reflecting light

limitation due to nearly permanent sea ice cover (Fig. 8a).

Earlier studies in Arctic Eurasia estuary sediments have reported $\delta^{13}C_{org}$ values ranging from -27.8‰

to -24.7‰ (Bröder et al., 2019; Tesi et al., 2014). Lowest values within our study area (-27.5‰ to -26‰,

Fig. 8b) are found in the western ESS shelf sediments receiving land-derived material from the Indigirka

and Lena rivers in agreement with previous data (-27.5‰ to -25.5‰, Bröder et al., 2019). Apart from

permafrost thawing, sea ice retreat likely accelerated coastal erosion contributing to the transfer of

allochthonous material towards the Arctic Ocean. The $\delta^{13}C_{org}$ values of organic matter produced by

phytoplankton in the Arctic Ocean vary around -24 ± 3‰ (Stein et al., 2004; Vonk et al., 2012), which

are similar to those found in saline waters of the western CS characterized by high pelagic production

(Fig. 8b). Enriched $\delta^{13}C_{org}$ values (-22‰ to -20.5‰) are also observed in the marginal ice zone/seasonal

sea-ice zone of the CS (Fig. 8b) where sea-ice plankton production is significant, which is in accordance

with values reported in sea ice (-23.6‰ to -18.3‰) (Schubert and Calvert 2001).

$\delta^{13}C_{org}$ and H-print were combined to discriminate among autochthonous and allochthonous sources

of OC in our surface sediments. As illustrated by the scatter plot in figure 8c, a three- end-member

distribution emerged from the relationship between these 2 parameters and summer sea ice concentration



(SuSIC) demonstrating their potential to differentiate sympagic, pelagic and terrestrial OC in our dataset. The site symbolizing the sympagic end-member is located at the sea-ice edge where sea-ice phytoplankton production is high. Coincidentally, its $\delta^{13}C_{org}$ value (-20.5‰) falls within the range of values reported in the lowermost 10 cm of sea ice cores (-23.6‰ to -18.3‰, Schubert and Calvert 2001) and close to the mean $\delta^{13}C_{org}$ value of sedimentary IP$_{25}$ (-19.3‰±2.3‰, Belt et al., 2008). The terrigenous

end-member is consistently represented by sites located in estuarine zones or where land-derived supply dominates. Pelagic end-member values (-23.5‰) are encountered in the western CS where phytoplankton productivity is the highest and the influence of terrigenous and sea ice carbon less pronounced. This more depleted value as compared to IP$_{25}$ is consistent with the latter being produced in sea ice. Indeed, the $\delta^{13}C_{org}$ of the two marine components also depends on $p\mathrm{CO}_2$. Higher $p\mathrm{CO}_2$ in surface

waters may result in lower $\delta^{13}C_{org}$ in pelagic marine algae OC while sea ice diatoms are relatively enriched in $\delta^{13}C_{org}$ due to potentially limited $\mathrm{CO}_2$ in sea ice (Tortell et al., 2013). Nevertheless, H-print, $\delta^{13}C_{org}$ and summer sea ice concentration (SuSIC) enabled us to successfully discriminate OC sources among our sediment sample set.

Based on these results, a ternary mixing model was used to calculate the relative contribution of

sympagic, pelagic, and terrestrial sources in our surface sediments (Fig. 8c and A5; Table B1) using the following equation:

$$\delta^{13}C_{sample} = f_{sym} \times \delta^{13}C_{sym} + f_{pela} \times \delta^{13}C_{pela} + f_{terr} \times \delta^{13}C_{terr} \qquad (4)$$

$$H-print_{sample} = f_{sym} \times H-print_{sym} + f_{pela} \times H-print_{pela} \qquad (5)$$

$$f_{sym} + f_{pela} + f_{terr} = 100\% \qquad (6)$$

where $f_{sym}$, $f_{pela}$, and $f_{terr}$ are the sympagic, pelagic, and terrestrial fractions of OC, respectively. The $\delta^{13}C_{org}$ end-members for used for the sympagic end-member at 0% for H-print is -20.5‰. The pelagic $\delta^{13}C_{org}$ end-member at 100% for H-print is -23.5‰. The terrestrial $\delta^{13}C_{org}$ end-member is -27.5‰.

**_5.2.2 Source and burial of organic carbon change over the last two centuries_**

In this section, we examine the temporal evolution of sympagic, pelagic and terrestrial fractions of OC calculated with our ternary mixing model and end-member values, with changing sea ice conditions over the last two centuries. Figure 9a shows that TOC values slowly increase between 1820 and 1860 and remain relatively stable around 0.8% until 1960 where they increase rapidly to reach their highest



levels (1.17%) at the core-top, falling in the range reported in the surface sediments of the CS shelf and

CP (1.25% to 2.56%, Goñi et al., 2013; 0.31% to 1.73%, Ji et al., 2019). This trend is paralleled by

increasing $\delta^{13}C_{org}$ values (Fig. 9a), suggesting that higher TOC may be related to enhanced marine

production (primary and secondary). Indeed, sea ice retreat and increase ice free conditions are expected

to result in higher rates of primary production due to higher light penetration and nutrients supply (both

form river and via wind driven mixing), a longer season of production and subsequently enhanced export

to the sea floor (Ouyang et al., 2022; Zhuang et al., 2022).

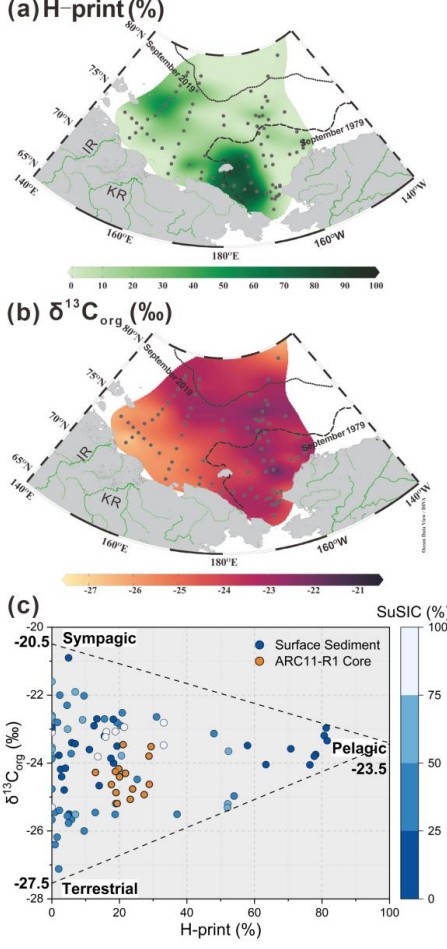

**Figure 8:** Distribution of (a) H-print and (b) $\delta^{13}C_{org}$ in the surface sediments of the ESS and CS. (c) Cross-plot of values of $\delta^{13}C_{org}$ and H-print for surface sediments. The gradient from dark blue to white represents SuSIC (Summer Sea Ice Concentration, NSIDC) from 0-20%, 20-40%, 40-60%, 60-80% and 80-100%, respectively, and ARC11-R1

core (in orange). The dotted and dashed lines in black represent the 20% isolines of September sea-ice concentration for the 2019 and 1979, respectively.




**Table 3.** Overall organic carbon composition of core ARC11-R1 (relative portion, $f_{oc}$ (%), and absolute content, OC (mg g$^{-1}$ d. w.)).

| Core depth | Age | $f_{sym}$ | $f_{pela}$ | $f_{terr}$ | OC$_{sym}$ | OC$_{pela}$ | OC$_{terr}$ |
|---|---|---|---|---|---|---|---|
| (cm) | (yr AD) | (%) | (%) | (%) | (mg g$^{-1}$ d. w.) | (mg g$^{-1}$ d. w.) | (mg g$^{-1}$ d. w.) |
| 0-1 | 2013 | 40.14 | 29.40 | 30.45 | 4.73 | 3.46 | 3.59 |
| 1-2 | 2000 | 36.50 | 28.77 | 34.72 | 4.12 | 3.24 | 3.92 |
| 2-3 | 1986 | 45.70 | 21.09 | 33.20 | 4.30 | 1.99 | 3.13 |
| 3-4 | 1972 | 32.77 | 20.05 | 47.18 | 3.10 | 1.90 | 4.47 |
| 4-5 | 1959 | 36.32 | 19.79 | 43.89 | 2.67 | 1.45 | 3.22 |
| 5-6 | 1945 | 35.68 | 18.83 | 45.49 | 2.75 | 1.45 | 3.50 |
| 6-7 | 1932 | 38.71 | 12.9 | 48.39 | 3.15 | 1.05 | 3.93 |
| 7-8 | 1918 | 33.23 | 21.83 | 44.94 | 2.57 | 1.69 | 3.48 |
| 8-9 | 1904 | 24.57 | 28.86 | 46.57 | 2.10 | 2.46 | 3.98 |
| 9-10 | 1891 | 30.89 | 17.64 | 51.47 | 2.47 | 1.42 | 4.14 |
| 10-11 | 1877 | 26.71 | 19.00 | 54.29 | 2.24 | 1.60 | 4.56 |
| 11-12 | 1864 | 25.42 | 23.91 | 50.67 | 2.18 | 2.05 | 4.35 |
| 12-13 | 1850 | 21.11 | 27.34 | 51.55 | 1.66 | 2.15 | 4.06 |
| 13-14 | 1837 | 21.96 | 19.34 | 58.70 | 1.59 | 1.40 | 4.26 |
| 14-15 | 1823 | 21.48 | 23.23 | 55.29 | 1.51 | 1.64 | 3.91 |

The C/N ratios are higher than the Redfield ratio over the entire core and show a decreasing trend

(Fig. 9b), in agreement with diminishing contribution of terrestrial with respect to marine OC sources.

The mean $f_{terr}$ is relatively high throughout the R1 core (46.45% ± 8.20%, Fig. 9c and Table 3) but

decrease from 58.70% to 30.45%. Although R1 is located rather far from the continent, these estimates

suggest efficient transport pathways of land-derived organic matter towards the open sea such as drifting

sea ice (Jia et al., 2019) or wave action remobilizing and transporting shelf sediments to the ocean interior

(Vonk et al., 2012). Time series sediment traps evidenced high and long-standing terrigenous material

advection by lateral transport controlled by Chukchi Slope Current and mesoscale eddies in winter in this

region (Onodera et al., 2021; Watanabe et al., 2014; Watanabe et al., 2022). Figure 9c also shows that

since the 1960s, there has been a significant decline in the share of terrigenous OC with the enhanced

loss of sea ice of the last 60 years, and subsequent rise of productivity of marine and sea ice diatoms. By

contrast, ARC4-C07, the core located to the southeast of R1 (Fig. 1), exhibits a trend of enhanced land-

sourced input in recent years (Bai et al., 2022). The different trend of the terrigenous OC components of

these two cores might be related to the weakening of sediment-laden sea ice transport to higher northern

latitudes due to its earlier melting.



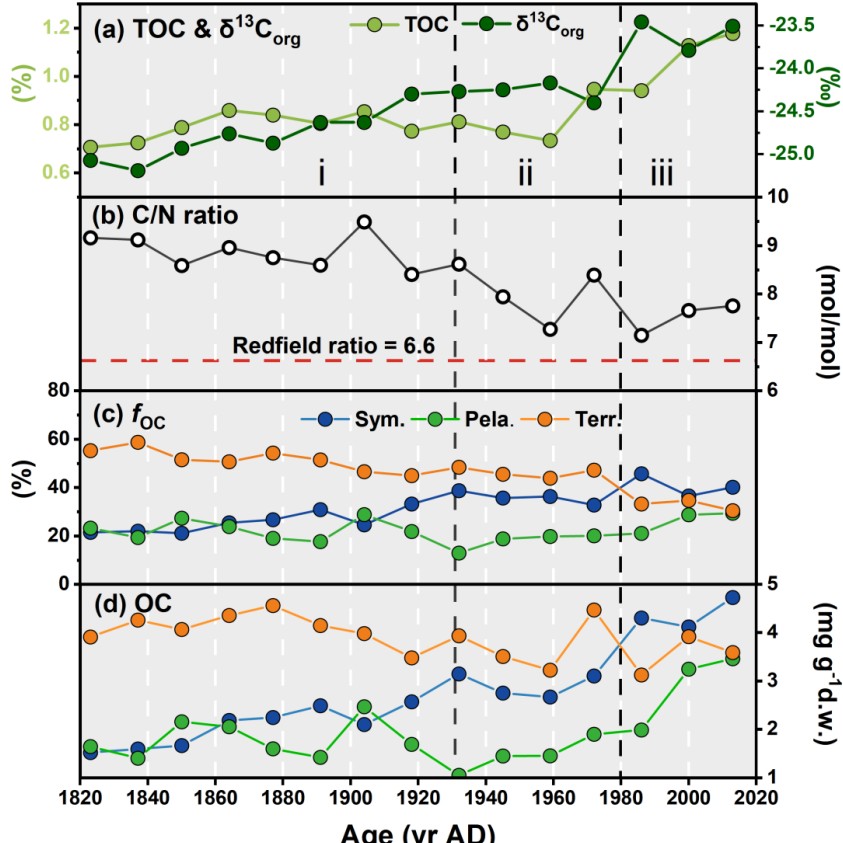

**Figure 9:** Downcore profiles of the concentrations of (a) TOC and $\delta^{13}C_{org}$, (b) C/N ratio, and (c) Proportion of overall

organic carbon from each source ($f_{OC}$, calculated based on $\delta^{13}C_{org}$ and H-print), and (d) OC (mg g$^{-1}$ d. w., dry weight)

in the ARC11-R1 core. In (c) and (d), blue, green, and orange dots represent the sympagic, pelagic and terrestrial

carbon input, respectively.

From the 1820s to 1930s, pelagic and sympagic OC are found in comparable amounts in the

sediments (2.17 ± 0.53 mg g$^{-1}$ d. w. sediment and 1.72 ± 0.46 mg g$^{-1}$ d. w. sediment, respectively; Fig.

9d and Table 3), but from the 1930s to 1980s, the content of sympagic OC increase by nearly 50% to

3.10 mg g$^{-1}$ d. w. sediment (Fig. 9d). However, pelagic phytoplankton growth also shows rising values

but remain lower than sympagic OC possibly because of surface freshening limiting nutrient supply

(Arrigo et al., 2008; Arrigo et al., 2012). After the 1980s, when the minimum ice edge reached R1 and

the ice-free period prolonged in the northern Chukchi Sea (Astakhov et al., 2019), the sympagic and

pelagic productions depict a final increase and superseding terrestrial OC (Fig. 9c and 9d). The high $\delta^{15}N$



data during this period suggest enhanced stratification most likely due to surface freshening (Fig. 3d) which gradually shifted the limiting factor of marine pelagic production from light to nutrients availability (Ardyna and Arrigo 2020; Lannuzel et al., 2020).

**6 Conclusions**

H-print and $\delta^{13}C_{org}$ values from 83 surface sediments were used to diagnose OC sources across the western Arctic Ocean and their link to SuSIC. $\delta^{13}C_{org}$ values were consistently generally lower in the coastal and shelf sediments where riverine terrigenous inputs accumulated while heavier $\delta^{13}C_{org}$ were found at offshore sites of the CS and ESS where pelagic and/or sympagic productions are significant. Combining $\delta^{13}C_{org}$ and H-print enabled to inventory terrestrial, sympagic and marine pelagic OC in our

sediment samples using a ternary mixing model which was then applied to the ARC11-R1 core to reconstruct their temporal evolution since 1820 as the sea ice retreat.

Our results demonstrate that over the last 200 years sea ice in the northern CS experienced nearly permanent sea ice conditions (1820s - 1830s) followed by a period of rapid melting to reach variable sea ice conditions persisting till the end of the 19th century. The onset of the 20th century was marked by

the return to icier conditions peaking in the 1930s followed by several decades of important melting (1930s -1980s) and the accelerated decline of sea ice in the most recent decades (1980s - present). These changing sea ice conditions led to shifts in primary production that, in turn, altered the composition and burial of OC in Arctic sediments from coastal to open sea areas. We show that with the loss of sea ice, the fraction of terrigenous OC decreased while marine pelagic and sympagic OC production gradually

increased leading to higher export and sequestration of marine OC in the deep-ocean. Since the beginning of the 21rst century the three OC pools at the core site are comparable, as opposed to 200 years ago when OC was predominantly of terrestrial origin. As temperature continues rising and sea ice melts, it is foreseeable that primary production in the northern CS may become nutrients limited, either due to freshening as it happens now, or wind driven vertical mixing as in open sea systems, and will undergo

shifts in phytoplankton populations and subsequently, $CO_2$ drawdown.





## Appendix A

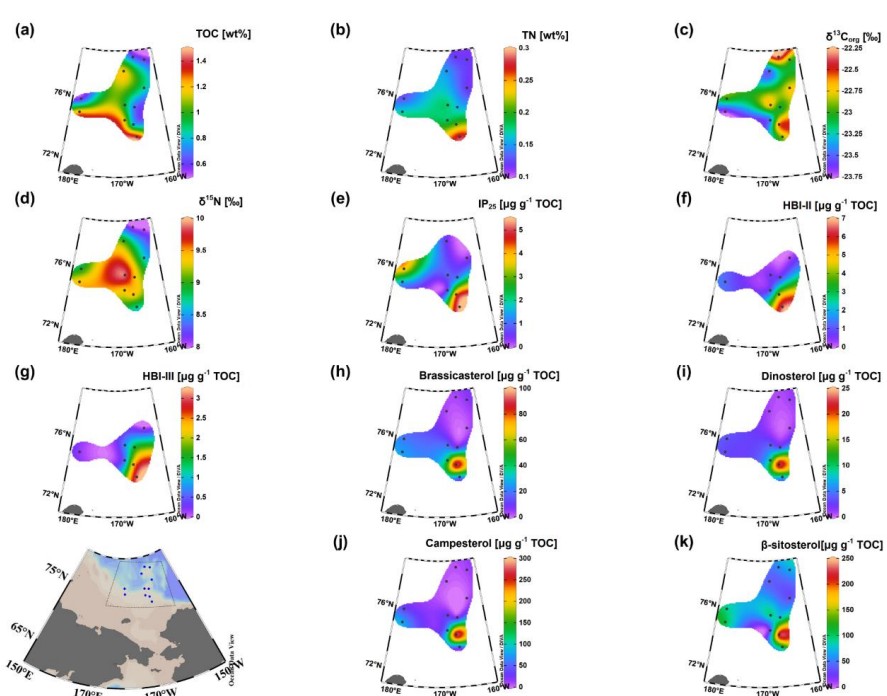

**Figure A1:** Distribution of (a) TOC, (b) TN, (c) $\delta^{13}C_{org}$, (d) $\delta^{15}N$, (e) $IP_{25}$, (f) HBI-II, (g) HBI-III, (h) brassicasterol, (i) dinosterol, (j) campesterol and (k) β-sitosterol in the surface sediments of Chukchi Sea and Chukchi Plateau.






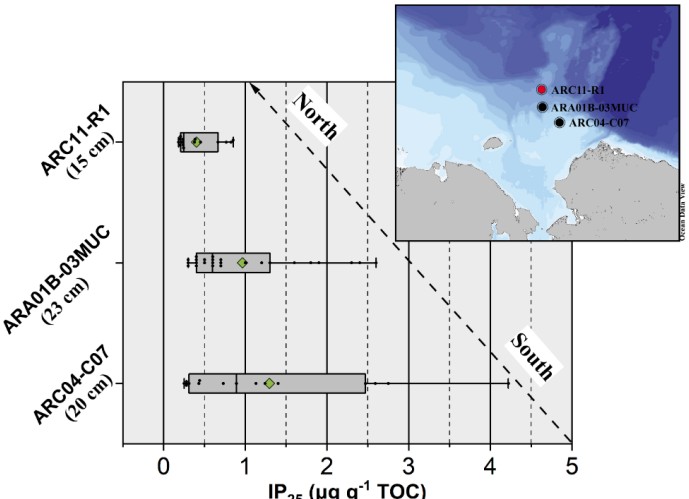

**Figure A2:** Box plot of IP$_{25}$ concentrations in 4 sediment cores (ARC04-C07, Bai et al. (2022); ARA01B-03MUC, Kim et al. (2019); ARC11-R1,). The map in the upper right shows their locations. The central bar in the boxes represents the median value and the green diamond represents the mean value. The rightmost and leftmost of the boxes represent the 75th and 25th percentiles, respectively. Whiskers are the maximum and minimum values within 1.5 times the interquartile range.




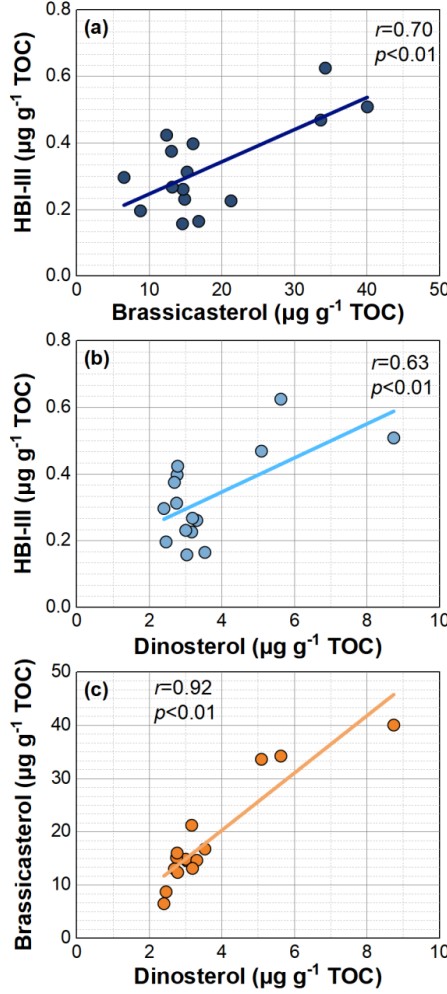

**Figure A3:** (a) Correlation between brassicasterol concentrations and HBI-III concentrations. (b) Correlation between dinosterol concentrations and HBI-III concentrations. (c) Correlation between dinosterol concentrations

and brassicasterol concentrations.

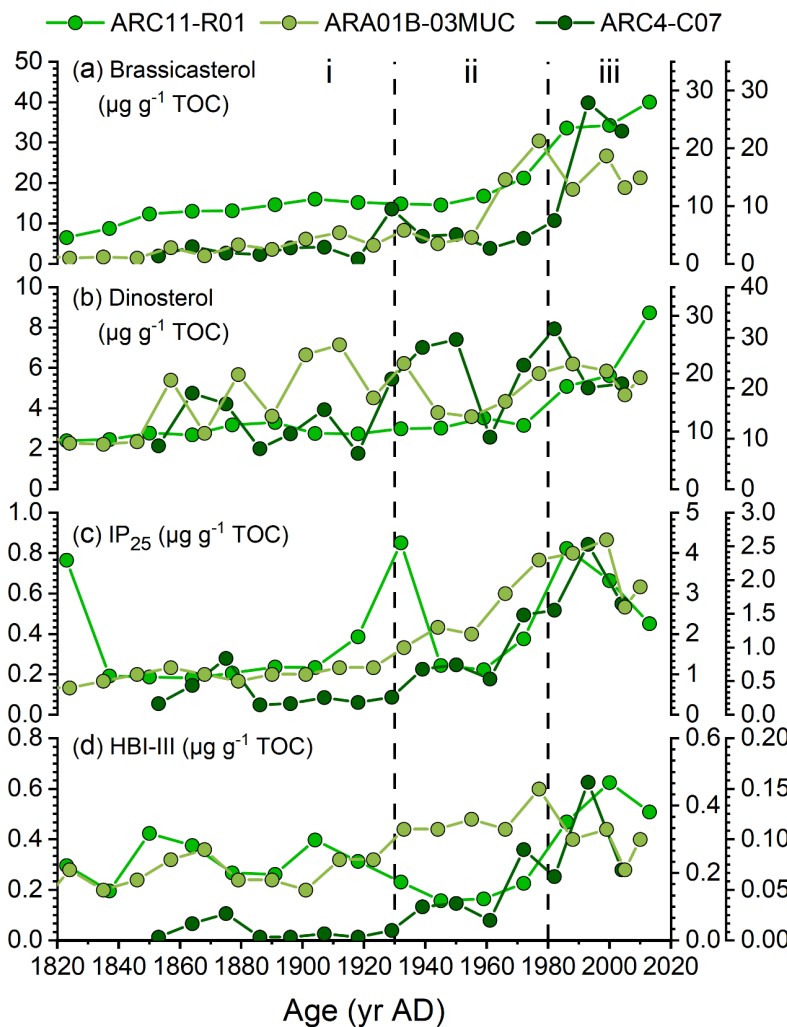

**Figure A4:** Downcore biomarker profiles of (a) brassicasterol concentration, (b) dinosterol concentration, (c) IP$_{25}$ concentration and (d) HBI-III concentration in the core ARC04-C07 (Bai et al. 2022), ARA01B-03MUC (Kim et al. 2019) and ARC11-R1.




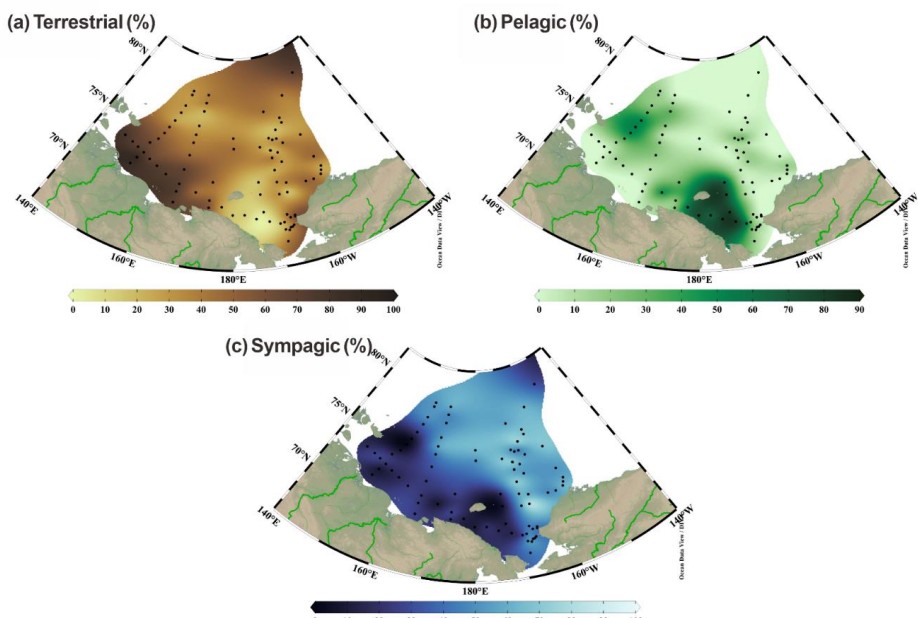

**Figure A5:** Proportion of organic carbon from each source in surface sediment: (a) terrestrial, (b) pelagic, and (c) sympagic.



**Appendix B**

**Table B1. Summary of TOC, H-print, $\delta^{13}C_{org}$ and proportion of overall organic carbon from each source ($f_{OC}$ (%), based on $\delta^{13}C_{org}$ and H-print, and OC (mg g$^{-1}$ d. w., dry weight)) data from surface sediment across the East Siberian Sea and the Chukchi Sea.**

| Cruise | Station | Longitude | Latitude | TOC (%) | H-print (%) | $\delta^{13}C_{org}$ (‰) | $f_{sym}$ (%) | $f_{pela}$ (%) | $f_{terr}$ (%) | OC$_{sym}$ (mg g$^{-1}$ d. w.) | OC$_{pela}$ (mg g$^{-1}$ d. w.) | OC$_{terr}$ (mg g$^{-1}$ d. w.) |
|---|---|---|---|---|---|---|---|---|---|---|---|---|
| LV77 | LV77-2 | -169.91 | 68.58 | 1.25 | 78.18 | -23.75 | 8.95 | 78.18 | 12.87 | 1.12 | 9.78 | 1.61 |
| LV77 | LV77-3 | -172.15 | 68.88 | 2.06 | 80.69 | -23.19 | 15.46 | 80.69 | 3.85 | 3.18 | 16.58 | 0.79 |
| LV77 | LV77-4 | -174.90 | 69.20 | 2.01 | 81.28 | -22.97 | 18.33 | 81.28 | 0.39 | 3.68 | 16.33 | 0.08 |
| LV77 | LV77-5 | -173.21 | 69.71 | 1.69 | 81.57 | -23.34 | 12.82 | 81.57 | 5.61 | 2.17 | 13.79 | 0.95 |
| LV77 | LV77-6 | -173.61 | 72.20 | 1.78 | 70.79 | -23.58 | 15.55 | 70.79 | 13.66 | 2.76 | 12.58 | 2.43 |
| LV77 | LV77-7 | -173.49 | 71.18 | 0.86 | 76.46 | -24.05 | 5.58 | 76.46 | 17.96 | 0.48 | 6.59 | 1.55 |
| LV77 | LV77-8 | -177.48 | 69.59 | 0.95 | 57.98 | -23.48 | 24.27 | 57.98 | 17.75 | 2.31 | 5.52 | 1.69 |
| LV77 | LV77-9 | 179.86 | 69.59 | 0.60 | 77.77 | -23.80 | 8.45 | 77.77 | 13.78 | 0.51 | 4.66 | 0.83 |
| LV77 | LV77-10 | 177.31 | 70.25 | 1.32 | 63.42 | -24.04 | 13.15 | 63.42 | 23.43 | 1.73 | 8.34 | 3.08 |
| LV77 | LV77-11 | 174.34 | 70.12 | 1.06 | 18.93 | -25.19 | 22.14 | 18.93 | 58.93 | 2.35 | 2.01 | 6.26 |
| LV77 | LV77-12 | 174.36 | 70.73 | 1.07 | 14.02 | -24.66 | 32.52 | 14.02 | 53.46 | 3.47 | 1.50 | 5.70 |
| LV77 | LV77-14 | 174.79 | 72.24 | 1.25 | 25.27 | -24.48 | 28.70 | 25.27 | 46.03 | 3.58 | 3.15 | 5.73 |
| LV77 | LV77-15 | 170.89 | 71.25 | 1.27 | 54.00 | -24.97 | 5.30 | 54.00 | 40.70 | 0.67 | 6.87 | 5.18 |
| LV77 | LV77-16 | 166.04 | 70.08 | 0.45 | 5.07 | -25.50 | 25.67 | 5.07 | 69.26 | 1.16 | 0.23 | 3.13 |
| LV77 | LV77-17 | 166.22 | 71.01 | 0.26 | 0.00 | -25.67 | 26.21 | 0.00 | 73.79 | 0.67 | 0.00 | 1.89 |
| LV77 | LV77-18 | 166.54 | 71.67 | 0.65 | 5.81 | -25.66 | 23.02 | 5.81 | 71.17 | 1.49 | 0.38 | 4.60 |
| LV77 | LV77-19 | 162.63 | 72.15 | 0.33 | 4.60 | -25.55 | 25.17 | 4.60 | 70.23 | 0.84 | 0.15 | 2.34 |
| LV77 | LV77-20 | 166.87 | 72.90 | 1.07 | 6.99 | -25.51 | 24.46 | 6.99 | 68.55 | 2.63 | 0.75 | 7.36 |
| LV77 | LV77-21 | 167.49 | 74.13 | 0.88 | 20.53 | -24.29 | 34.13 | 20.53 | 45.34 | 3.02 | 1.82 | 4.01 |
| LV77 | LV77-22 | 167.83 | 75.18 | 0.96 | 20.60 | -23.78 | 41.34 | 20.60 | 38.06 | 3.96 | 1.97 | 3.65 |
| LV77 | LV77-23 | 168.10 | 75.85 | 0.96 | 4.67 | -23.00 | 61.63 | 4.67 | 33.70 | 5.89 | 0.45 | 3.22 |
| LV77 | LV77-24 | 168.51 | 76.60 | 0.62 | 21.40 | -22.94 | 52.91 | 21.40 | 25.69 | 3.27 | 1.32 | 1.59 |
| LV77 | LV77-25 | 169.25 | 77.81 | 0.51 | 33.17 | -22.81 | 48.02 | 33.17 | 18.81 | 2.43 | 1.68 | 0.95 |
| LV77 | LV77-26 | 169.55 | 78.49 | 0.57 | 13.63 | -23.81 | 44.93 | 13.63 | 41.44 | 2.54 | 0.77 | 2.34 |
| LV77 | LV77-27 | 169.76 | 79.15 | 1.63 | 0.00 | -23.05 | 63.59 | 0.00 | 36.41 | 10.34 | 0.00 | 5.92 |
| LV77 | LV77-28 | 163.49 | 79.19 | 0.41 | 15.90 | -23.23 | 51.99 | 15.90 | 32.11 | 2.16 | 0.66 | 1.33 |
| LV77 | LV77-29 | 163.28 | 78.85 | 0.66 | 18.82 | -23.07 | 52.50 | 18.82 | 28.68 | 3.44 | 1.23 | 1.88 |
| LV77 | LV77-30 | 162.05 | 77.90 | 0.62 | 16.17 | -23.08 | 53.85 | 16.17 | 29.98 | 3.34 | 1.00 | 1.86 |
| LV77 | LV77-31 | 161.31 | 77.24 | 0.32 | 33.09 | -23.47 | 38.66 | 33.09 | 28.25 | 1.22 | 1.04 | 0.89 |
| LV77 | LV77-32 | 160.25 | 76.50 | 0.80 | 52.33 | -23.65 | 25.11 | 52.33 | 22.56 | 2.02 | 4.21 | 1.81 |
| LV77 | LV77-33 | 159.26 | 75.85 | 0.94 | 52.12 | -25.20 | 3.09 | 52.12 | 44.79 | 0.29 | 4.89 | 4.20 |
| LV77 | LV77-34 | 158.50 | 75.25 | 0.77 | 52.05 | -25.31 | 1.56 | 52.05 | 46.39 | 0.12 | 4.03 | 3.59 |
| LV77 | LV77-35 | 157.42 | 74.64 | 0.88 | 37.10 | -25.51 | 7.29 | 37.10 | 55.61 | 0.64 | 3.27 | 4.90 |
| LV77 | LV77-36 | 155.65 | 74.10 | 0.92 | 12.84 | -25.46 | 21.86 | 12.84 | 65.30 | 2.00 | 1.18 | 5.98 |
| LV77 | LV77-38 | 159.77 | 72.56 | 0.68 | 9.56 | -26.00 | 15.99 | 9.56 | 74.45 | 1.09 | 0.65 | 5.05 |





| Cruise | Station | Longitude | Latitude | TOC (%) | H-print (%) | $\delta^{13}C_{org}$ (‰) | $f_{sym}$ (%) | $f_{pela}$ (%) | $f_{terr}$ (%) | $OC_{sym}$ (mg g$^{-1}$ d. w.) | $OC_{pela}$ (mg g$^{-1}$ d. w.) | $OC_{terr}$ (mg g$^{-1}$ d. w.) |
|---|---|---|---|---|---|---|---|---|---|---|---|---|
| LV77 | LV77-39 | 157.34 | 72.87 | 0.37 | 18.75 | -25.51 | 17.71 | 18.75 | 63.54 | 0.65 | 0.69 | 2.33 |
| LV77 | LV77-40 | 153.25 | 71.90 | 0.96 | 0.93 | -26.40 | 15.18 | 0.93 | 83.89 | 1.46 | 0.09 | 8.08 |
| LV77 | LV77-41 | 154.13 | 72.55 | 0.75 | 2.02 | -27.12 | 4.35 | 2.02 | 93.63 | 0.33 | 0.15 | 7.05 |
| LV77 | LV77-42 | 155.19 | 73.16 | 1.22 | 4.95 | -25.77 | 21.94 | 4.95 | 73.11 | 2.67 | 0.60 | 8.88 |
| LV77 | LV77-43 | 153.16 | 73.37 | 0.78 | 0.91 | -25.45 | 28.77 | 0.91 | 70.32 | 2.23 | 0.07 | 5.45 |
| LV77 | LV77-44 | 151.19 | 73.54 | 0.60 | 2.21 | -25.39 | 28.85 | 2.21 | 68.94 | 1.72 | 0.13 | 4.10 |
| LV77 | LV77-45 | 148.50 | 73.70 | 0.69 | 1.56 | -26.18 | 17.97 | 1.56 | 80.47 | 1.24 | 0.11 | 5.54 |
| ARC11 | 20Z4 | -166.61 | 73.54 | 1.78 | 20.31 | -22.89 | 54.25 | 20.31 | 25.44 | 9.63 | 3.61 | 4.52 |
| ARC11 | 20Z3 | -167.16 | 74.34 | 0.78 | 20.85 | -22.52 | 59.24 | 20.85 | 19.91 | 4.64 | 1.63 | 1.56 |
| ARC11 | 20P1-6 | -166.62 | 75.44 | 0.69 | 48.04 | -23.13 | 34.98 | 48.04 | 16.98 | 2.40 | 3.30 | 1.16 |
| ARC11 | 20P2-5 | -163.68 | 76.60 | 0.87 | 9.09 | -22.73 | 62.99 | 9.09 | 27.92 | 5.45 | 0.79 | 2.42 |
| ARC11 | 20E2 | 179.99 | 75.84 | 0.57 | 0.00 | -22.99 | 64.44 | 0.00 | 35.56 | 3.66 | 0.00 | 2.02 |
| ARC11 | 20E1 | -179.89 | 75.01 | 1.23 | 5.70 | -23.72 | 50.73 | 5.70 | 43.57 | 6.25 | 0.70 | 5.37 |
| ARC11 | 20R2 | -168.92 | 75.61 | 0.96 | 31.02 | -22.64 | 51.70 | 31.02 | 17.28 | 4.96 | 2.97 | 1.66 |
| ARC11 | 20R1 | -169.13 | 74.64 | 0.76 | 9.51 | -22.02 | 72.85 | 9.51 | 17.64 | 5.53 | 0.72 | 1.34 |
| ARC11 | 20R5 | -168.94 | 77.76 | 1.18 | 0.00 | -23.54 | 56.63 | 0.00 | 43.37 | 6.66 | 0.00 | 5.10 |
| ARC06 | 14R14 | -160.43 | 78.63 | 0.45 | 0.00 | -23.10 | 62.86 | 0.00 | 37.14 | 2.83 | 0.00 | 1.67 |
| ARC06 | 14R12 | -163.89 | 77.00 | 0.50 | 0.00 | -22.70 | 68.57 | 0.00 | 31.43 | 3.43 | 0.00 | 1.57 |
| ARC06 | 14R11 | -166.20 | 76.15 | 0.79 | 0.00 | -21.60 | 84.29 | 0.00 | 15.71 | 6.66 | 0.00 | 1.24 |
| ARC06 | 14R10 | -167.90 | 75.43 | 0.56 | 0.00 | -23.80 | 52.86 | 0.00 | 47.14 | 2.96 | 0.00 | 2.64 |
| ARC06 | 14R09 | -169.03 | 74.61 | 1.33 | 6.90 | -21.90 | 76.06 | 6.90 | 17.04 | 10.12 | 0.92 | 2.27 |
| ARC06 | 14R08 | -169.00 | 74.00 | 1.27 | 4.49 | -22.30 | 71.72 | 4.49 | 23.79 | 9.11 | 0.57 | 3.02 |
| ARC06 | 14R07 | -168.97 | 73.00 | 1.47 | 2.34 | -22.00 | 77.23 | 2.34 | 20.43 | 11.35 | 0.34 | 3.00 |
| ARC06 | 14R03 | -169.05 | 68.62 | 1.11 | 13.33 | -22.70 | 60.95 | 13.33 | 25.72 | 6.77 | 1.48 | 2.85 |
| ARC06 | 14S03 | -157.08 | 72.24 | 1.75 | 19.24 | -22.80 | 56.15 | 19.24 | 24.61 | 9.83 | 3.37 | 4.31 |
| ARC06 | 14S02 | -157.46 | 71.92 | 1.72 | 15.58 | -22.70 | 59.67 | 15.58 | 24.75 | 10.26 | 2.68 | 4.26 |
| ARC06 | 14S01 | -157.93 | 71.62 | 1.07 | 1.67 | -22.80 | 66.19 | 1.67 | 32.14 | 7.08 | 0.18 | 3.44 |
| ARC06 | 14C04 | -166.99 | 71.01 | 1.28 | 4.94 | -20.90 | 91.46 | 4.94 | 3.60 | 11.71 | 0.63 | 0.46 |
| ARC06 | 14C01 | -168.14 | 69.22 | 0.89 | 11.11 | -23.60 | 49.37 | 11.11 | 39.52 | 4.39 | 0.99 | 3.52 |
| ARC06 | 14C03 | -166.48 | 69.03 | 1.06 | 2.90 | -24.40 | 42.63 | 2.90 | 54.47 | 4.52 | 0.31 | 5.77 |
| ARC06 | 14C13-5 | -159.18 | 75.20 | 0.75 | 0.00 | -22.60 | 70.00 | 0.00 | 30.00 | 5.25 | 0.00 | 2.25 |
| ARC06 | 14CC6 | -167.13 | 68.24 | 0.60 | 18.31 | -24.00 | 39.54 | 18.31 | 42.15 | 2.37 | 1.10 | 2.53 |
| ARC06 | 14CC4 | -167.51 | 68.13 | 0.58 | 19.30 | -23.50 | 46.11 | 19.30 | 34.59 | 2.67 | 1.12 | 2.01 |
| ARC06 | 14CC3 | -167.90 | 68.10 | 0.38 | 12.50 | -22.90 | 58.57 | 12.50 | 28.93 | 2.23 | 0.48 | 1.10 |
| ARC06 | 14CC2 | -168.24 | 67.90 | 0.53 | 15.56 | -23.20 | 52.54 | 15.56 | 31.90 | 2.78 | 0.82 | 1.69 |
| ARC03 | 08R15 | -169.01 | 73.99 | 1.18 | 0.00 | -24.76 | 39.14 | 0.00 | 60.86 | 4.62 | 0.00 | 7.18 |
| ARC03 | 08R11 | -168.98 | 72.00 | 1.71 | 0.00 | -22.94 | 65.14 | 0.00 | 34.86 | 11.14 | 0.00 | 5.96 |
| ARC03 | 08R09 | -168.97 | 70.99 | 1.33 | 6.90 | -23.43 | 54.20 | 6.90 | 38.90 | 7.21 | 0.92 | 5.17 |
| ARC03 | 08R03 | -169.02 | 68.00 | 1.68 | 18.18 | -22.71 | 58.04 | 18.18 | 23.78 | 9.75 | 3.05 | 4.00 |



| Cruise | Station | Longitude | Latitude | TOC (%) | H-print (%) | $\delta^{13}C_{org}$ (‰) | $f_{sym}$ (%) | $f_{pela}$ (%) | $f_{terr}$ (%) | $OC_{sym}$ (mg g$^{-1}$ d. w.) | $OC_{pela}$ (mg g$^{-1}$ d. w.) | $OC_{terr}$ (mg g$^{-1}$ d. w.) |
|--------|---------|-----------|----------|---------|-------------|--------------------------|---------------|----------------|----------------|--------------------------------|---------------------------------|---------------------------------|
| ARC03 | 08R01 | -169.00 | 67.00 | 0.73 | 17.50 | -23.86 | 42.00 | 17.50 | 40.50 | 3.07 | 1.28 | 2.96 |
| ARC03 | 08M07 | -171.99 | 75.01 | 1.11 | 0.00 | -22.64 | 69.43 | 0.00 | 30.57 | 7.71 | 0.00 | 3.39 |
| ARC03 | 08S14 | -157.92 | 73.17 | 1.27 | 0.00 | -23.87 | 51.86 | 0.00 | 48.14 | 6.59 | 0.00 | 6.11 |
| ARC03 | 08B11 | -165.03 | 75.00 | 1.18 | 0.00 | -23.60 | 55.71 | 0.00 | 44.29 | 6.57 | 0.00 | 5.23 |
| ARC03 | 08C33 | -167.51 | 68.92 | 1.17 | 4.00 | -24.15 | 45.57 | 4.00 | 50.43 | 5.33 | 0.47 | 5.90 |
| ARC03 | 08C35 | -166.51 | 68.92 | 1.55 | 3.64 | -24.79 | 36.63 | 3.64 | 59.73 | 5.68 | 0.56 | 9.26 |
| ARC03 | 08C13 | -166.75 | 71.80 | 1.45 | 5.89 | -23.40 | 55.21 | 5.89 | 38.90 | 8.01 | 0.85 | 5.64 |
| ARC03 | 08C17 | -161.98 | 71.49 | 1.49 | 2.58 | -24.18 | 45.95 | 2.58 | 51.47 | 6.85 | 0.38 | 7.67 |
| ARC03 | 08C19 | -159.98 | 71.45 | 1.19 | 2.09 | -23.77 | 52.09 | 2.09 | 45.82 | 6.20 | 0.25 | 5.45 |



**Data availability**

All data that support the findings of this study are included within the article and appendices.

**Author contributions**

L.S., J.R. and J.C. designed the study and wrote the manuscript with contribution of M.-A.S., Y.B., Z.L.,

R. Z., H.J., A.A.S. and X.S. L.S. and R. Z. contributed the biomarker analyses and the determination of

bulk parameter. J.R. retrieved the environmental data from different database while X.H. carried out the

age model estimate of R1. All authors contributed to the final version of the manuscript.

**Competing interests:** The authors declare that they have no conflict of interest.

**Acknowledgements**

We are grateful to the captain, crewmembers and scientific party of the R/V *Xuelong 2* and R/V *Akademik*

*M.A. Lavrentiev* for their professional sampling work. We are indebted to Zhi Yang and Qianna Chen of

Second Institute of Oceanography for their kind help in the bulk data measurement. Vincent Klein of

Sorbonne University is also thanked for technical assistance. We are also grateful to Sabine Schmidt of

University of Bordeaux for her suggestions on the age model. This study is financially supported by the

National Natural Science Foundation of China (Nos. 41941013, 42076241, 41976229, 41606052,

42076242), the National Key Research and Development Program of China (Nos. 2019YFE0120900,

2019YFC1509101) and the Marine S&T Fund of Shandong Province for Pilot National Laboratory for

Marine Science and Technology (Qingdao; No. 2018SDKJ0104-3). The expedition work was partly

supported by the Ministry of Sciences and Education of the Russian Federation (Project 121021700342-

9).



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
