# Peer review of "Changing sources and burial of organic carbon in the Chukchi Sea sediments with retreating sea ice over recent centuries"

_EGUsphere, 2023_

## Author Comment (AC1)

The manuscript "Changing sources and burial of organic carbon in the Chukchi Sea sediments with retreating sea ice over recent centuries" by Su et al. describes the response of the organic matter in Chukchi Sea sediments to arctic environmental changes based on the biomarkers in their surface sediments and sediment core. Overall, their manuscript is generally well written and organized. Methods seem to be appropriate. Results are illustrated with relevant Graphs and Tables. Discussion is focused on the main findings and comprehensive. However, I would encourage the authors to slightly improve their manuscript according to my suggestions as follows:

We appreciate the constructive review of our manuscript and addressed comments thereafter in blue.

Line 251-252: "The presence of IP$_{25}$ throughout R1 indicates that sea ice cover has been seasonal at least since the 1820s at this location"- As IP$_{25}$ can also be detected at sea ice edge (Müller et al., 2011), and under sea ice blooms have been reported (Arrigo et al., 2012), is it possible that the IP$_{25}$ in R1 comes from the sympagic algae from sea ice edge, rather than from seasonal sea ice? Besides, this is contrary to the conclusion of permeant sea ice in line 288-289. This part needs further discussion.

Seasonal sea ice comprises sea ice formation in winter and melting during summer. IP$_{25}$ produced in sea ice is exported during melting. To our knowledge sea ice edge production of IP$_{25}$ has not been strictly demonstrated, and thus in absence of evidence, these productions are undistinguishable. Nevertheless, we slightly re-phrased this sentence but tuned it to a more general statement. Under sea ice bloom (UIB) has been reported by Arrigo et al. (2012) and Coupel et al. (2012, 2015), but to our knowledge it is however unrelated to the production of IP$_{25}$.

We agree that Lines 288-289 requires further discussion which has now been incorporated in the revised manuscript.

Arrigo, K.R., Perovich, D.K., Pickart, R.S., Brown, Z.W., van Dijken, G.L., Lowry, K.E., Mills, M.M., Palmer, M.A., Balch, W.M., Bahr, F., Bates, N.R., Benitez-Nelson, C., Bowler, B.,

*Brownlee, E., Ehn, J.K., Frey, K.E., Garley, R., Laney, S.R., Lubelczyk, L., Mathis, J., Matsuoka, A., Mitchell, B.G., Moore, G.W.K., Ortega-Retuerta, E., Pal, S., Polashenski, C.M., Reynolds, R.A., Schieber, B., Sosik, H.M., Stephens, M. and Swift, J.H.: Massive Phytoplankton Blooms Under Arctic Sea Ice, Science, 336(6087), 1408-1408, https://doi.org/10.1126/science.1215065, 2012.*

*Coupel, P., Jin, H.Y., Joo, M., Horner, R., Bouvet, H.A., Sicre, M.A., Gascard, J.C., Chen, J.F., Garçon, V. and Ruiz-Pino, D.: Phytoplankton distribution in unusually low sea ice cover over the Pacific Arctic, Biogeosciences, 9, 4835–4850, 2012.*

*Coupel, P., Ruiz-Pino, D., Sicre, M.A., Chen, J.F., Lee, S.H., Schiffrine, N., Li, H.L. and Gascard, J.C.: The impact of freshening on phytoplankton production in the Pacific Arctic Ocean, Prog. Oceanogr., 131, 113-125, https://doi.org/ 10.1016/j.pocean.2014.12.003, 2015.*

Line 282-283: "This result further confirm that HBI-III producers proliferate at the sea ice edge rather than in ice free waters."- Is it possible the low HBI-III is a result of limited nutrient which was deleted by the blooms, rather than different producers?

This is a good question. However, the results from the time-series sediment trap at the Northwind Ridge, western Arctic Ocean (see the figure below; Bai et al., 2019) suggest

that the low values of HBI-III in summer are unlikely to be primarily caused by nutrient depletion, because during this period, brassicasterol is high, indicating favorable nutrient conditions for phytoplankton blooming, which precludes nutrient depletion.

Bai, Y., Sicre, M.-A., Chen, J., Klein, V., Jin, H., Ren, J., Li, H., Xue, B., Ji, Z., Zhuang, Y., and Zhao, M.: Seasonal and spatial variability of sea ice and phytoplankton biomarker flux in the Chukchi sea (western Arctic Ocean), Prog. Oceanogr., 171, 22–37, https://doi.org/10.1016/j.pocean.2018.12.002, 2019.

Line 379-380: "higher light penetration and nutrients supply (both form river and via wind driven mixing)" – do you mean river and mixing will result in higher light penetration and much more nutrient?

This part of the discussion is on riverine input and wind driven mixing as sources/drivers of nutrient supply, not light penetration. We revised the sentence to avoid misleading. Primary production is controlled by light and nutrients. Subsequently sea ice retreat results in higher light penetration triggering higher rates of primary production. As sea ice cover reduces, nutrients supply from rivers can spread in larger areas. In addition, wind driven mixing can exert stress in the upper ocean and replete surface layers with nutrients from deeper layers. Finally, longer ice free season as well contribute to enhance production and export to the sea floor (Ouyang et al., 2022; Zhuang et al., 2022).

Ouyang, Z., Li, Y., Qi, D., Zhong, W., Murata, A., Nishino, S., Wu, Y., Jin, M., Kirchman, D., Chen, L., and Cai, W.-J.: The Changing $CO_2$ Sink in the Western Arctic Ocean From 1994 to 2019, Global Biogeochem. Cycles, 36, https://doi.org/10.1029/2021gb007032, 2022.

Zhuang, Y., Jin, H., Cai, W.-J., Li, H., Qi, D., and Chen, J.: Extreme Nitrate Deficits in the Western Arctic Ocean: Origin, Decadal Changes, and Implications for Denitrification on a Polar Marginal Shelf, Global Biogeochem. Cycles, 36, e2022GB007304, https://doi.org/10.1029/2022GB007304, 2022.

Line 437-440: Why would wind driven vertical mixing lead to nutrient limitation? Why would these changes in phytoplankton lead to $CO_2$ drawdown?

The sentence has been rephrased. Surface freshening and thus enhanced summer

stratification as in open sea systems, is what was meant here. Changes in phytoplankton as for example from micro- to nano-plankton has implication of the uptake of $CO_2$ and export.

Line 296: spring/summer sea ice – should be summer/fall sea ice as the sea ice in this area starts to melt in summer (line 124 "receding as the summer season (July) begins") It is hard to define a uniform seasonality in the study region covering from 67°N to 80°N, as the southern area may experience summer, meanwhile the northern part enters spring. $IP_{25}$ is a proxy for seasonal sea ice, therefore the concentration of sea ice is the key information.

- In Fig. 7d ii) and iii), why was the winter sea ice edge (in March) in the Chukchi Sea (ARC04-C07), rather that the Bering Sea? And why was there abundance HBI-III (+) under the permanent sea ice cover in Fig. 7d i)?

Thanks for pointing out this error, we have corrected Figure 7 and amended the corresponding sentence in the revised manuscript. Under sea ice phytoplankton blooms (UIB) were evidenced in the Arctic Ocean (e.g. Arrigo et al., 2012, 2014; Coupel et al., 2012, 2015). It is possible that the production of HBI-III under permanent sea ice was produced from UIBs, which has been recently evidenced by Gal et al (2022). Also, biomarker signals transported by sea ice and currents cannot be ignored.

Arrigo, K.R., Perovich, D.K., Pickart, R.S., Brown, Z.W., van Dijken, G.L., Lowry, K.E., Mills, M.M., Palmer, M.A., Balch, W.M., Bahr, F, Bates, N.R., Benitez-Nelson, C., Bowler, B., Brownlee, E., Ehn, J.K., Frey, K.E., Garley, R., Laney, S.R., Lubelczyk, L., Mathis, J., Matsuoka, A., Mitchell, B.G., Moore, G.W.K., Ortega-Retuerta, E., Pal, S., Polashenski, C.M., Reynolds, R.A., Schieber, B., Sosik, H.M., Stephens, M. and Swift, J.H.: Massive Phytoplankton Blooms Under Arctic Sea Ice, Science, 336(6087), 1408-1408, https://doi.org/10.1126/science.1215065, 2012.
Arrigo, K. R., Perovich, D. K., Pickart, R. S., Brown, Z. W., van Dijken, G. L., Lowry, K. E., Mills, M. M., Palmer, M. A., Balch, W. M., Bates, N. R., Benitez-Nelson, C. R., Brownlee, E., Frey, K. E., Laney, S. R., Mathis, J., Matsuoka, A., Greg Mitchell, B., Moore, G. W. K., Reynolds, R. A., Sosik, H. M., and Swift, J. H.: Phytoplankton blooms beneath the sea ice in the Chukchi sea, Deep Sea Res., Part II, 105, 1–16, https://doi.org/10.1016/j.dsr2.2014.03.018, 2014.

*Coupel, P., Jin, H.Y., Joo, M., Horner, R., Bouvet, H.A., Sicre, M.A., Gascard, J.C., Chen, J.F., Garçon, V. and Ruiz-Pino, D.: Phytoplankton distribution in unusually low sea ice cover over the Pacific Arctic, Biogeosciences, 9, 4835–4850, 2012.*

*Coupel, P., Ruiz-Pino, D., Sicre, M.A., Chen, J.F., Lee, S.H., Schiffrine, N., Li, H.L. and Gascard, J.C.: The impact of freshening on phytoplankton production in the Pacific Arctic Ocean, Prog. Oceanogr., 131, 113-125, https://doi.org/ 10.1016/j.pocean.2014.12.003, 2015.*

*Gal, J.-K., Ha, S.-Y., Park, J., Shin, K.-H., Kim, D., Kim, N.-Y., Kang, S.-H., and Yang, E. J.: Seasonal Flux of Ice-Related Organic Matter During Under-Ice Blooms in the Western Arctic Ocean Revealed by Algal Lipid Biomarkers, J. Geophys. Res.: Oceans, 127, https://doi.org/10.1029/2021jc017914, 2022.*

Line 26:"δ" should be italic throughout the text.

Done.

Line 28:"were also" should be "are also".

Done.

Line 32:summer sea ice

Done.

Line 120:"is summer" should be "in summer"

Done.

Line 120: "seasonal" should be deleted.
Done.

Line 103-104: "The dynamics of the Beaufort Gyre (BG) also impacts on the characteristics of the CS water mass." - provide references here.

We've added *Timmermans and Toole, 2023* as the reference.

*Timmermans, M.-L. and Toole, J. M.: The Arctic Ocean's Beaufort Gyre, Annu. Rev. Mar. Sci., 15, 223–248, https://doi.org/10.1146/annurev-marine-032122-012034, 2023.*

Line 107-109: "This basin is connected to the Pacific Ocean through…"- sentence needs to be restructured

Thanks. We revised the sentence accordingly.

Line 123-125: "Remote sensing data evidence strong seasonal variations, with sea…"- awkward sentence

Done. We revised the sentence to "Remote sensing data (1979 to 2020) reveal considerable seasonal variations of sea ice extent in the CS (Cavalieri et al., 1996). The CS is heavily covered by sea ice from November to June. Sea ice gradually decreases in July and reaches its minimum extent in September."

*Cavalieri D J, Parkinson C L, Gloersen P, Zwally H J. Sea Ice Concentrations from Nimbus-7 SMMR and DMSP SSM/I-SSMIS Passive Microwave Data, Version 1. Boulder, Colorado USA: NASA National Snow and Ice Data Center Distributed Active Archive Center (Digital media, updated yearly), 1996.*

Line 139: "excess $^{210}$Pb ($^{210}$Pbex)" should be put in line 137

Done.

Line 178: 3β-ol)).

Done.

Line 181: What is "cholesterol-d6"? Do you mean "cholest-5-en-3β-ol-D6"?

Revised.

Line 192: "The H-print values were also calculated to infer the…"- should come before formula (3)

Done.

Line 202: Please using R1 for ARC11-R1 in the text after line 130.

We have double checked the entire manuscript and have used R1 consistently.

Line 221: varies

Done.

Line 232: present

Done.

Line 236: why "they"?

Done.

Line 238: spans

Corrected.

Line 239-240: - sentence needs to be restructured.

We fully revised the sentence.

Line 244: "our core" should be instead by "R01" throughout the text.

Done.

Line 250: lower export of sympagic OC

Added.

Line 268: found in North of Iceland

Added.

Line 292: Wu et al. (2019)

Done.

Line 333: freshwater discharge as lowering salinity water can suppress HBI-III production.

Revised to *…by lowering water salinity…*

Line 337: within our study area – should be deleted.

Done.

Line 339-341: "Apart from permafrost thawing, sea ice retreat likely accelerated coastal erosion…" - provide references here

We've added *Overeem et al., 2011* as the reference.

*Overeem, I., Anderson, R. S., Wobus, C. W., Clow, G. D., Urban, F. E., and Matell, N.: Sea ice loss enhances wave action at the Arctic coast, Geophys. Res. Lett., 38, https://doi.org/10.1029/2011GL048681, 2011.*

Line 353: Schubert and Calvert, 2001

Corrected.

Line 362: The abbreviation of SuSIC has already appeared in the previous paragraph.

Corrected.

Line 365: Fig. A5

Corrected.

Line 404: "the concentrations of" should be deleted.

Done.

Line 414: northern CS

Done.

Line 418: Ardyna and Arrigo, 2020

Corrected.

- "Figure" or "Fig." please be consistent throughout the text (but the figure caption).

- $CO_2$ with "2" in lower case, please check it throughout the text including references.

- "Pacific water inflow" or "Pacific Water Inflow", "Siberian coastal current" or "Siberian Coastal Current", "11th" or "11th", "sea ice edge" or "sea-ice edge", "Figure" or "Fig." -please keep these be consistent throughout the text.

We are very grateful to the reviewer for the thorough and deep examination of our manuscript. We have carefully examined the manuscript to rectify and harmonize the wordings.

---

## Author Comment (AC2)

Liang Su and Co-Authors present a new proxy downcore record from the Chukchi Sea covering the past 200 years combined with results from surface sediments. The aim is to investigate different sources (pelagic/sympagic) of organic carbon in the study area. With their record they show the relationship between sea ice and organic matter input to better the understanding of the mechanisms driving the marine carbon cycle in the Arctic Ocean. This study is relevant due to the ongoing and expected climate changes in the Arctic Ocean especially regarding the fate of organic carbon produced, delivered in the Chukchi Sea, an area of dramatic sea-ice loss in the recent years.

The manuscript is well written and the presented work is of importance to the scientific community. However, some points need clarification and/or correction before publication.

Apologies, in case I have misinterpreted anything.

We greatly appreciate the referee in reviewing our manuscript. Please refer to the one-to-one response below in blue for specific modifications and clarifications.

**General comments:**

The interpretation of the $IP_{25}/PIP_{25}$ record needs some attention. I think the biggest weakness is, that you do not use your $IP_{25}$ record from surface sediments to verify the interpretation of your down core record, e.g., to validate the $PIP_{25}$ index and its environmental signal.

This is a good point. We have made sensitivity test on $c$-factor based on the surface sediments from the study region. See the detailed answers to specific comments below.

Further an age control of surface sediments should at least be discussed. Surface sediments may not always represent modern conditions or a mix of several hundred to thousand years, which makes the comparison with satellite data from a very specific time interval difficult. Further details on this are given in the Specific Comments.

Limitation linked to the age of the surface sediments is indeed an issue and we have

added a discussion on this in the revised manuscript. For a detailed explanation see the responses to specific comments below.

In the figures, the time on the x-axis is displayed from left/old to right/young. In my understanding, it is common to show old ages on the right and young ages on the left in the palaeoceanographic community. This may also be the reason for some inconsistencies in the order results are described, see comment below.

The PAGES2k community working on the Common Era climate (last 2000 years) has agreed to have Present on the right, while paleoceanographers working on more ancient climate have it on the left (cf papers of Pages2k network; PAGES2k, 2013, 2019).

*PAGES2k Consortium. 2013. Continental-scale temperature variability during the past two millennia. Nature Geoscience, 6, 339–346.*

*PAGES2k Consortium. 2019. Consistent multidecadal variability in global temperature reconstructions and simulations over the Common Era. Nature Geoscience, 12, 643-649.*

**Specific Comments**

**Introduction**

**L79** Belt et al., 2007, de Vernal et al., 2013 seem rather old and rather specific proxy studies. Further there have been updates on the given references and many other studies regarding this topic.

We updated the references and selected review articles (Belt, 2018; de Vernal et al., 2013) for their broader coverage.

*Belt, S. T.: Source-specific biomarkers as proxies for Arctic and Antarctic sea ice, Org. Geochem., 125, 277–298, https://doi.org/10.1016/j.orggeochem.2018.10.002, 2018.*

*de Vernal, A., Gersonde, R., Goosse, H., Seidenkrantz, M.-S., and Wolff, E. W.: Sea ice in the paleoclimate system: the challenge of reconstructing sea ice from proxies – an introduction, Quat. Sci. Rev., 79, 1–8, https://doi.org/10.1016/j.quascirev.2013.08.009, 2013.*

**Fig 1** The blue dots are barely visible, please change the color. The black pentagram is too small. What is the source of the sea ice margins, please add a reference.

We changed the blue dots to red ones and enlarged the star now in red to make it more visible. References of the sea ice margins were added in the figure caption ([https://nsidc.org](https://nsidc.org), Cavalieri et al., 1996).

*Cavalieri D J, Parkinson C L, Gloersen P, Zwally H J. Sea Ice Concentrations from Nimbus-7 SMMR and DMSP SSM/I-SSMIS Passive Microwave Data, Version 1. Boulder, Colorado USA: NASA National Snow and Ice Data Center Distributed Active Archive Center (Digital media, updated yearly), 1996.*

**Oceanographic Setting**

**L123-125** What is the time interval of the sea-ice dataset you used?

We have used satellite sea ice concentration data from 1979 to 2020 for seasonal of sea ice. This is now indicated in the revised manuscript.

**Results**

**L231-235** One time you describe your records from old to young, the other time, from young to old. Please be consistent.

We have carefully checked the description of the results and harmonized it.

**L238** $IP_{25}$ concentrations span

Done.

**Material & Methods**

**L128** In what year where the surface sediments taken? Have they all been measured to represent modern sediments? This is relatively important when comparing them to modern sea-ice concentrations.

This has been the approach taken by paleoceanographer for years, without even using box cores but core-tops that most the time were not modern. We are fully aware of the limitation of this approach and added a sentence in the revised manuscript (see 5.2.1). Nevertheless, distinct end-numbers were resolved from the ternary model diagram. Therefore, we believe that our method is still reliable.

**L196-199** Why do you exclude other surface records, e.g., Wegner-Koch et al., 2020.

Koch et al. (2020) analyzed biomarkers in surface sediments from the Bering Sea and Chukchi Sea, which is indeed in our research area. However, Koch et al. (2020) deals with H-print data but not $\delta^{13}C_{org}$, data, which has limited its use for this study.

**Discussion**

**L246-252** Yes, the concentrations are lower in the southern cores from Bai et al. and Kim et al. What is missing here is a discussion about potential differences in core storage, method as mentioned by Belt et al. (2014, Clim. Past). Are there other factors that may limit productivity, nutrients, depositional system?

We agree and re-emphasized the effect of storage and methods which is well-known within the proxy community.

L258 Cabedo-Sanz et al., 2013 worked in Barents Sea. I would recommend to mention that, as you adapt their interpretation to a new area.

This section has been removed as not only it is not in the same area but it related to a different time period (Younger Dyas).

**L270 ff**

I see a problem here with the interpretation of the PIP$_{25}$ index.

- It does not make sense to use c-balance factors from other studies and other areas. As mentioned by previous surface studies Xiao et al., Kolling et al., a balance factor should be calculated based on data as there are differences in the concentrations of individual biomarkers varying between regions, (not to mention geological time intervals) depending on extraction method, storage etc... All of these make it not valid to use balance factors from surface sediments from Barents Sea (Smik et al., 2016) and a dataset from the Nordic Seas & Arctic Ocean (Xiao et al., 2015a). Further, Xiao et al.(2015a) used a different extraction method. I advise to calculate

a balance factor based on your surface and downcore data, which is also recommended by Xiao et al.

| $c$ / Age | Surface sediment (this study) | | | R1 core (this study) | | | Xiao et al. (2015) & Smik et al. (2016) | | |
|---|---|---|---|---|---|---|---|---|---|
| | *1.29* | *0.03* | *0.13* | *1.23* | *0.02* | *0.11* | *0.63* | *0.02* | *0.11* |
| | $P_{III}IP_{25}$ | $P_BIP_{25}$ | $P_DIP_{25}$ | $P_{III}IP_{25}$ | $P_BIP_{25}$ | $P_DIP_{25}$ | $P_{III}IP_{25}$ | $P_BIP_{25}$ | $P_DIP_{25}$ |
| 2013 | 0.41 | 0.27 | 0.28 | 0.42 | 0.36 | 0.32 | 0.58 | 0.36 | 0.32 |
| 2000 | 0.45 | 0.39 | 0.48 | 0.46 | 0.49 | 0.52 | 0.63 | 0.49 | 0.52 |
| 1986 | 0.58 | 0.45 | 0.55 | 0.59 | 0.55 | 0.60 | 0.74 | 0.55 | 0.60 |
| 1972 | 0.56 | 0.37 | 0.48 | 0.57 | 0.47 | 0.52 | 0.73 | 0.47 | 0.52 |
| 1959 | 0.51 | 0.31 | 0.33 | 0.52 | 0.40 | 0.37 | 0.68 | 0.40 | 0.37 |
| 1945 | 0.55 | 0.36 | 0.38 | 0.56 | 0.46 | 0.42 | 0.71 | 0.46 | 0.42 |
| 1932 | 0.74 | 0.66 | 0.69 | 0.75 | 0.74 | 0.72 | 0.85 | 0.74 | 0.72 |
| 1918 | 0.49 | 0.46 | 0.52 | 0.50 | 0.56 | 0.56 | 0.66 | 0.56 | 0.56 |
| 1904 | 0.31 | 0.33 | 0.39 | 0.32 | 0.42 | 0.44 | 0.48 | 0.42 | 0.44 |
| 1891 | 0.41 | 0.35 | 0.36 | 0.42 | 0.45 | 0.39 | 0.59 | 0.45 | 0.39 |
| 1877 | 0.38 | 0.34 | 0.33 | 0.39 | 0.44 | 0.37 | 0.55 | 0.44 | 0.37 |
| 1864 | 0.27 | 0.32 | 0.34 | 0.28 | 0.41 | 0.38 | 0.43 | 0.41 | 0.38 |
| 1850 | 0.26 | 0.34 | 0.34 | 0.27 | 0.43 | 0.38 | 0.41 | 0.43 | 0.38 |
| 1837 | 0.43 | 0.42 | 0.38 | 0.44 | 0.52 | 0.42 | 0.61 | 0.52 | 0.42 |
| 1823 | 0.67 | 0.80 | 0.71 | 0.68 | 0.85 | 0.74 | 0.80 | 0.85 | 0.74 |

In order to address these issues we did sensitivity tests on *c*-factors to evaluate their effect on sea ice reconstructions using Xiao et al. (2015) and Smik et al. (2016) versus dataset of this study (our surface sediment set, and R1 core). The table above and figures below shows *c*-factors calculated for different cases and their corresponding $PIP_{25}$ values. Differences between $P_BIP_{25}$ and $P_DIP_{25}$ values based on various *c*-factors are minor, as compared to the limitation of these indexes under high to permanent sea ice conditions (19[th] and early 20[th] centuries; Walsh et al., 2019). $P_{III}IP_{25}$ values based on *c*-factors from our surface sediments and R1 core are slightly lower than those derived from Smik et al. (2016) but with similar fluctuations through the record (see the figure below). Aslo, Kim et al. (2019) studied the same region and suggested that the $PIP_{25}$ derived sea ice reconstructions were more reliable by using *c*-factors from the pan-

Arctic database (Xiao et al. (2015) and Smik et al. (2016)). Therefore we keep using the *c*-factors from Xiao et al. (2015) and Smik et al. (2016) in this study.

Kim, J.H., Gal, J.K., Jun, S.Y., Smik, L., Kim, D., Belt, S.T., Park, K., Shin, K.H. and Nam, S.I.:. *Reconstructing spring sea ice concentration in the Chukchi Sea over recent centuries: insights into the application of the PIP25 index. Environ. Res. Lett., 14, 125004, 2019.*

Smik, L., Cabedo-Sanz, P., and Belt, S. T.: *Semi-quantitative estimates of paleo Arctic sea ice concentration based on source-specific highly branched isoprenoid alkenes: a further development of the PIP$_{25}$ index, Org. Geochem., 92, 63–69, https://doi.org/10.1016/j.orggeochem.2015.12.007, 2016.*

Xiao, X., Fahl, K., Müller, J., and Stein, R.: *Sea-ice distribution in the modern Arctic Ocean: Biomarker records from trans-Arctic Ocean surface sediments, Geochim. Cosmochim. Acta, 155, 16–29, https://doi.org/10.1016/j.gca.2015.01.029, 2015.*

[Figure]

Xiao et al., 2015 and Smik et al., 2016

[Figure]

Surface sediment (CS, n=47)

[Figure]

R1 core

- You are using the calibration from Müller et al (2012) from Fram Strait to interpret your $PIP_{25}$ results as percentages of sea ice cover. This is not correct. The calibration by Müller et al (2012) was done for Fram Strait which has not been validated for any other area, which is why Müller et al. and other surface studies (Xiao et al., 2015a, Kolling et al., 2020) recommend that this calibration is only roughly applicable for other regions. Hence, I would recommend that you do not use percentages in your interpretation but the general sea ice regime as introduced by Müller et al., 2012, e.g., ice-free, variable, permanent, which you also show in Fig 5

We now use numerical values and a more qualitative narrative.

**Fig 6** How do you define those margins for the different sea ice conditions? Are they based on your surface dataset, or any other published surface dataset? Or are they just estimates?

From my understanding below permanent sea ice, there should be no production of any biomarker, but you allow production of e.g., 20 mg/g TOC of Brassicasterol below permanent sea ice when maximum Brassicasterol concentrations are at 40 mg/g TOC. This does not seem realistic. In Fig 7 is becomes obvious that you barely have any $PIP_{25}$ values that indicate permanent sea ice cover, however in Fig 6 is seems as if at least half of your datapoints lie within the range that indicates permanent sea ice cover.

We used the Chukchi Sea surface sediment dataset to determine thresholds for sea ice conditions. The maximum concentrations of brassicasterol of the southern surface sediments are up to 100 µg g$^{-1}$ TOC, while these values decrease to less than 20 µg g$^{-1}$ TOC in the north where minimum sea ice extent is located (see the figure below). In contrast, the highest concentration of brassicasterol in the R1 core is about 40 µg g$^{-1}$ TOC with a mean of ~18 µg g$^{-1}$ TOC (Table 2). Therefore, it is reasonable to set a threshold of brassicasterol of 20 µg g$^{-1}$ TOC.

The phytoplankton biomarkers under ice cover may have been introduced by sea ice and currents, which is supported by contemporaneous terrestrial input signals ($\delta^{13}C_{org}$

and terrestrial sterols; Fig. 3 and 4).

[Figure]

**L285** Even though I am not a native speaker I feel that the wording 'icier' is not the correct scientific term. You use it several times in your manuscript. I would suggest using 'increase in sea ice' or something similar.

We have used a different wording than icier, "…sea ice increase or extended ice cover..".

**Fig 7a** It is really hard to distinguish the different shades of green, especially between ARC-11-R01 and ARC4-C07. Please use different colors.

We have modified the colors in Figure 7 to make them more visible.

**Fig7b** What is the effect of light availability in your record? In my understanding, there should be no production below winter sea ice due to the lack of sunlight in Chukchi Sea. Further, what is the main bloom season on Chukchi Sea? Please elaborate more on this topic.

Due to the lack of sunlight there is almost no primary production under the sea ice in winter, this has been corrected on the cartoons. The primary production process in the

Chukchi Sea begins in the south during spring, coinciding with the melting of the sea ice. As the sea ice gradually recedes towards the north, the hotspots of productivity also shift northwards. This process of primary production continues throughout the summer and does not complete until late autumn, when the sea ice freezes completely.

**L334** Is there a difference in light availability between ~70°N and ~80°N that could also influence the amount of biomarkers produced. How long are the general production periods in your working area, are there specific differences from North to South?

Obviously, there are differences in light availability from south to north of the study region. Due to the timing of sea ice melting and freezing in the western Arctic Ocean, the duration of primary production is significantly different between the north and south. For instance, the southern Chukchi region (~67.78°N) experienced an average open water duration of 152.3 ± 13.1 days from 1988 to 2018, compared to 109.7 ± 38.4 days in the north (72.16°N), thereby extending the production season by 3 weeks in the southern region. As a result, the net primary productivity in the south was 25.8% higher than that in the north (Payne et al., 2022). However, besides light availability, biomarker production is also influenced by nutrient supply, which is in turn driven by other factors such as stratification, water mass transport, terrestrial input, etc. These factors are also different between the south and the north of the study region.

*Payne, C. M., Dijken, G. L. van, and Arrigo, K. R.: North-South Differences in Under-Ice Primary Production in the Chukchi Sea From 1988 to 2018, J. Geophys. Res: Oceans, 127, e2022JC018431, https://doi.org/10.1029/2022JC018431, 2022.*

**L351 & L355** 'sea-ice edge' and 'sea ice carbon', if you hyphenate be consistent throughout the manuscript.

We have carefully revised the manuscript to harmonize.

**Fig 8** I would suggest that you write an endmember on the scales for H-print and $^{13}$C, e.g., sympagic/pelagic.

We highlighted the end-member in the figure caption.

**L392-395** If land-derived organic matter is transported to the core location by sea ice, why aren't IP$_{25}$ and terrigenous sterols not parallel? Could you include the biomarker records to this discussion?

Sedimentary sympagic/pelagic organic carbon were mixed signals from local production and long/near transport, whereas terrigenous material was only transported from long distances, so they do not necessarily parallel each other especially if this mechanism is marginal.

**Conclusions**

**L438** nutrient limited

The typo has been corrected.

**L440** CO$_2$ drawdown

Corrected.

---

## Author Response (AR2)

Dear editor Nathalie Combourieu Nebout,

We thank you for your constructive comments on our manuscript entitled "Changing sources and burial of organic carbon in the Chukchi Sea sediments with retreating sea ice over recent centuries" (No. egusphere-2023-64). We have revised the manuscript according to your suggestions. Besides, some minor mistakes or typos have been corrected. The Appendix Table and Figures have been put into the Supplement, according to the editing system of Climate of the Past. Therefore their corresponding names have been transferred to Table S1 and Figure S1-S6. Please find below our responses in **blue**.

Best regards,
Jian Ren (on behalf of all co-authors)
* * *
Nevertheless make attention to the bibliography. I have not seen some of references in the text. Please verify all of them and especially those below and then add in the text or remove from the bibliography.

Belt et al., 2019
Arrigo et al, 2012
Cabedo-Sanz et al., 2013 (if it is in the text, it remains bad-palced in the bibliography)
Müller and Stein, 2014
Sicre et al., 2001 and 2013
Stein et al., 2016
van Dongen et al, 2008
Zhang et al., 2012.

- We have removed these uncited references from the bibliography.

If it is possible, I would like an enlargement of the characters in the figure 8 a and b and in Appendix A1 and A6 to be more readable. The characters are really very small at 100%. In Appendix A2, please place the label on the cores on the left and they will be more readable.

- We have enlarged the characters in Figure 8 and Appendix A1 and A6 (now Figure S1 and S6). The labels in Appendix A2 have been placed to the left of the cores (now Figure S2).